# Training Structured Neural Networks Through Manifold Identification and Variance Reduction

**Zih-Syuan Huang**
Academia Sinica
zihsyuan@stat.sinica.edu.tw

**Ching-pei Lee**
Academia Sinica
leechingpei@gmail.com

## Abstract

This paper proposes an algorithm, RMDA, for training neural networks (NNs) with a regularization term for promoting desired structures. RMDA does not incur computation additional to proximal SGD with momentum, and achieves variance reduction without requiring the objective function to be of the finite-sum form. Through the tool of manifold identification from nonlinear optimization, we prove that after a finite number of iterations, all iterates of RMDA possess a desired structure identical to that induced by the regularizer at the stationary point of asymptotic convergence, even in the presence of engineering tricks like data augmentation that complicate the training process. Experiments on training NNs with structured sparsity confirm that variance reduction is necessary for such an identification, and show that RMDA thus significantly outperforms existing methods for this task. For unstructured sparsity, RMDA also outperforms a state-of-the-art pruning method, validating the benefits of training structured NNs through regularization. Implementation of RMDA is available at https://www.github.com/zihsyuan1214/rmda.

## 1 Introduction

Training neural networks (NNs) with regularization to obtain a certain desired structure such as structured sparsity or discrete-valued parameters is a problem of increasing interest. Existing approaches either use stochastic subgradients of the regularized objective (Wen et al., 2016; 2018) or combine popular stochastic gradient algorithms for NNs, like SGD with momentum (MSGD) or Adam (Kingma & Ba, 2015), with the proximal operator associated with the regularizer to conduct proximal stochastic gradient updates to obtain a model with preferred structures (Bai et al., 2019; Yang et al., 2019; Yun et al., 2021; Deleu & Bengio, 2021). Such methods come with proven convergence for certain measures of first-order optimality and have shown some empirical success in applications. However, we notice that an essential theoretical support lacking in existing methods is the guarantee for the output iterate to possess the same structure as that at the point of convergence. More specifically, often the imposed regularization is only known to induce a desired structure exactly at optimal or stationary points of the underlying optimization problem (see for example, Zhao & Yu, 2006), but training algorithms are only able to generate iterates asymptotically converging to a stationary point. Without further theoretical guarantees, it is unknown whether the output iterate, which is just an approximation of the stationary point, still has the same structure. For example, let us assume that sparsity is desired, the point of convergence is $x^* = (1, 0, 0)$, and two algorithms respectively produce iterates $\{y^t = (1, t^{-1}, t^{-1})\}$ and $\{z^t = (1 + t^{-1}, 0, 0)\}$. Clearly, both iterate sequences converge to $x^*$, but only $z^t$ has the same desired structure as its limit point $x^*$, while $y^t$ is not useful for sparsity despite that the point of convergence is. This work aims at filling this gap to propose an algorithm for training structured NNs that can provably make all its iterates after

a finite number of iterations possess the desired structure of the stationary point to which the iterates converge. We term the structure at a stationary point a stationary structure, and it should be understood that for multiple stationary points, each might correspond to a different stationary structure, and we aim at identifying the one at the limit point of the iterates of an algorithm, instead of selecting the optimal one among all stationary structures. Although finding the structure at an inferior stationary point might seem not very meaningful, another reason for studying this identification property is that for the same point of convergence, the structure at the limit point is the most preferable one. Consider the same example above, we note that for any sequence $\{x^t\}$ converging to $x^*$, $x_1^t \neq 0$ for all $t$ large enough, for otherwise $x^t$ does not converge to $x^*$. Therefore, $x^t$ cannot be sparser than $x^*$ if $x^t \to x^*$.[1] Identifying the structure of the point of convergence thus also amounts to finding the locally most ideal structure under the same convergence premise.

It is well-known in the literature of nonlinear optimization that generating iterates consistently possessing the structure at the stationary point of convergence is possible if all points with the same structure near the stationary point can be presented locally as a manifold along which the regularizer is smooth. This manifold is often termed as the active manifold (relative to the given stationary point), and the task of generating iterates staying in the active manifold relative to the point of convergence after finite iterations is called manifold identification (Lewis, 2002; Hare & Lewis, 2004; Lewis & Zhang, 2013). To identify the active manifold of a stationary point, we need the regularizer to be partly smooth (Lewis, 2002; Hare & Lewis, 2004) at that point, roughly meaning that the regularizer is smooth along the active manifold around the point, while the change in its value is drastic along directions leaving the manifold. A more technical definition will be given in Section 3. Fortunately, most regularizers used in machine learning are partly smooth, so stationary structure identification is possible, and various deterministic algorithms are known to achieve so (Hare & Lewis, 2007; Hare, 2011; Wright, 2012; Liang et al., 2017a;b; Li et al., 2020; Lee, 2020; Bareilles et al., 2020).

On the other hand, for stochastic gradient-related methods to identify a stationary structure, existing theory suggests that the variance of the gradient estimation needs to vanish as the iterates approach a stationary point (Poon et al., 2018), and indeed, it is observed empirically that proximal stochastic gradient descent (SGD) is incapable of manifold identification due to the presence of the variance in the gradient estimation (Lee & Wright, 2012; Sun et al., 2019).[2] Poon et al. (2018) showed that variance-reduction methods such as SVRG (Johnson & Zhang, 2013; Xiao & Zhang, 2014) and SAGA (Defazio et al., 2014) that utilize the finite-sum structure of empirical risk minimization to drive the variance of their gradient estimators to zero are suitable for this task. Unfortunately, with the standard practice of data augmentation in deep learning, training of deep learning models with a regularizer should be treated as the following stochastic optimization problem that minimizes the expected loss over a distribution, instead of the commonly seen finite-sum form:

$$\min_{W \in \mathcal{E}} \quad F(W) \coloneqq \mathbb{E}_{\xi \sim \mathcal{D}}\left[f_\xi(W)\right] + \psi(W), \tag{1}$$

where $\mathcal{E}$ is a Euclidean space with inner product $\langle \cdot, \cdot \rangle$ and the associated norm $\|\cdot\|$, $\mathcal{D}$ is a distribution over a space $\Omega$, $f_\xi$ is differentiable almost everywhere for all $\xi \in \Omega$, and $\psi(W)$ is a regularizer that might be nondifferentiable. We will also use the notation $f(W) \coloneqq \mathbb{E}_{\xi \sim \mathcal{D}}[f_\xi(W)]$. Without a finite-sum structure in (1), Defazio & Bottou (2019) pointed out that classical variance-reduction methods are ineffective for deep learning, and one major reason is the necessity of periodically evaluating $\nabla f(W)$ (or at least using a large batch from $\mathcal{D}$ to get a precise approximation of it) in variance-reduction methods is intractable, hence manifold identification and therefore finding the stationary structure becomes an extremely tough task for deep learning. Although recently there are efforts in developing variance-reduction methods for (1) inspired by online problems (Wang et al., 2019; Nguyen et al., 2021; Pham et al., 2020; Cutkosky & Orabona, 2019; Tran-Dinh et al., 2019), these methods all have multiple hyperparameters to tune and incur computational cost at least twice or

---

[1]See a more detailed discussion in Appendix B.1.

[2]An exception is the interpolation case, in which the variance of plain SGD vanishes asymptotically. But data augmentation often fails this interpolation condition.

thrice to that of (proximal) SGD. As the training of deep learning models is time- and resource-consuming, these drawbacks make such methods less ideal for deep learning.

To tackle these difficulties, we extend the recently proposed modernized dual averaging framework (Jelassi & Defazio, 2020) to the regularized setting by incorporating proximal operations, and obtain a new algorithm RMDA (Regularized Modernized Dual Averaging) for (1). The proposed algorithm provably achieves variance reduction beyond finite-sum problems without any cost or hard-to-tune hyperparameters additional to those of proximal momentum SGD (proxMSGD), and we provide theoretical guarantees for its convergence and ability for manifold identification. The key difference between RMDA and the original regularized dual averaging (RDA) of Xiao (2010) is that RMDA incorporates momentum and can achieve better performance for deep learning in terms of the generalization ability, and the new algorithm requires nontrivial proofs for its guarantees. We further conduct experiments on training deep learning models with a regularizer for structured-sparsity to demonstrate the ability of RMDA to identify the stationary structure without sacrificing the prediction accuracy.

When the desired structure is (unstructured) sparsity, a popular approach is pruning that trims a given dense model to a specified level, and works like (Gale et al., 2019; Blalock et al., 2020; Evci et al., 2020; Verma & Pesquet, 2021) have shown promising results. However, as a post-processing approach, pruning is essentially different from structured training considered in this work, because pruning is mainly used when a model is available, while structured training combines training and structure inducing in one procedure to potentially reduce the computational cost and memory footprint when resources are scarce. We will also show in our experiment that RMDA can achieve better performance than a state-of-the-art pruning method, suggesting that structured training indeed has its merits for obtaining sparse NNs.

The main contributions of this work are summarized as follows.

- ***Principled analysis***: We use the theory of manifold identification from nonlinear optimization to provide a unified way towards better understanding of algorithms for training structured neural networks.

- ***Variance reduction beyond finite-sum with low cost***: RMDA achieves variance reduction for problems that consist of an infinite-sum term plus a regularizer (see Lemma 2) while incorporating momentum to improve the generalization performance. Its spatial and computational cost is almost the same as proxMSGD, and there is no additional hyperparameters to tune, making RMDA suitable for large-scale deep learning.

- ***Structure identification***: With the help of variance reduction, our theory shows that under suitable conditions, after a finite number of iterations, iterates of RMDA stay in the active manifold of its limit point.

- ***Superior empirical performance***: Experiments on neural networks with structured sparsity exemplify that RMDA can identify a stationary structure without reducing the validation accuracy, thus outperforming existing methods by achieving higher group sparsity. Another experiment on unstructured sparsity also shows RMDA outperforms a state-of-the-art pruning method.

After this work is finished, we found a very recent paper Kungurtsev & Shikhman (2021) that proposed the same algorithm (with slightly differences in the parameters setting in Line 5 of Algorithm 1) and analyzed the expected convergence of (1) under a specific scheduling of $c_t = s_{t+1}\alpha_{t+1}^{-1}$ when both terms are convex. In contrast, our work focuses on nonconvex deep learning problems, and especially on the manifold identification aspect.

## 2 ALGORITHM

Details of the proposed RMDA are in Algorithm 1. At the $t$-th iteration with the iterate $W^{t-1}$, we draw an independent sample $\xi_t \sim \mathcal{D}$ to compute the stochastic gradient $\nabla f_{\xi_t}(W^{t-1})$, decide a learning rate $\eta_t$, and update the weighted sum $V_t$ of previous stochastic gradients using $\eta_t$ and the scaling factor $\beta_t := \sqrt{t}$:

$$V_0 := 0, \quad V_t := \sum_{k=1}^{t} \eta_k \beta_k \nabla f_{\xi_k}(W^{k-1}) = V_{t-1} + \eta_t \beta_t \nabla f_{\xi_t}(W^{t-1}), \quad \forall t > 0.$$

**Algorithm 1:** RMDA $(W^0, T, \eta(\cdot), c(\cdot))$

---

**input** : Initial point $W^0$, learning rate schedule $\eta(\cdot)$, momentum schedule $c(\cdot)$,
        number of epochs $T$

1   $V_0 \leftarrow 0, \quad \alpha_0 \leftarrow 0$

2   **for** $t = 1, \ldots, T$ **do**

3     $\beta_t \leftarrow \sqrt{t}, \quad s_t \leftarrow \eta(t)\beta_t, \quad \alpha_t \leftarrow \alpha_{t-1} + s_t$

4     Sample $\xi_t \sim \mathcal{D}$ and compute $V^t \leftarrow V^{t-1} + s_t \nabla f_{\xi_t}(W^{t-1})$

5     $\tilde{W}^t \leftarrow \arg\min_W \langle V^t, W \rangle + \frac{\beta_t}{2}\|W - W^0\|^2 + \alpha_t \psi(W)$             // (2)

6     $W^t \leftarrow (1 - c(t))W^{t-1} + c(t)\tilde{W}^t$

**output:** The final model $W^T$

---

The tentative iterate $\tilde{W}^t$ is then obtained by the proximal operation associated with $\psi$:

$$\tilde{W}^t = \text{prox}_{\alpha_t \beta_t^{-1} \psi}\left(W^0 - \beta_t^{-1} V^t\right), \quad \alpha_t := \sum_{k=1}^t \beta_k \eta_k, \tag{2}$$

where for any function $g$, $\text{prox}_g(x) := \arg\min_y \|y - x\|^2/2 + g(y)$ is its proximal operator. The iterate is then updated along the direction $\tilde{W}^t - W^{t-1}$ with a factor of $c_t \in [0, 1]$:

$$W^t = (1 - c_t)W^{t-1} + c_t \tilde{W}^t = W^{t-1} + c_t\left(\tilde{W}^t - W^{t-1}\right). \tag{3}$$

When $\psi \equiv 0$, RMDA reduces to the modernized dual averaging algorithm of Jelassi & Defazio (2020), in which case it has been shown that mixing $W^{t-1}$ and $\tilde{W}^t$ in (3) equals to introducing momentum (Jelassi & Defazio, 2020; Tao et al., 2018). We found that this introduction of momentum greatly improves the performance of RMDA and is therefore essential for applying it on deep learning problems.

## 3 ANALYSIS

We provide theoretical analysis of the proposed RMDA in this section. Our analysis shows variance reduction in RMDA and stationarity of the limit point of its iterates, but all of them revolves around our main purpose of identification of a stationary structure within a finite number of iterations. The key tools for this end are partial smoothness and manifold identification (Hare & Lewis, 2004; Lewis, 2002). Our result is the currently missing cornerstone for those proximal algorithms applied to deep learning problems for identifying desired structures. In fact, it is actually well-known in convex optimization that those algorithms based on plain proximal stochastic gradient without variance reduction are *unable to identify the active manifold*, and the structure of the iterates oscillates due to the variance in the gradient estimation; see, for example, experiments and discussions in Lee & Wright (2012); Sun et al. (2019). Our work is therefore the first one to provide justification for solving the regularized optimization problem in deep learning to really identify a desired structure induced by the regularizer. Throughout, $\nabla f_\xi$ denotes the gradient of $f_\xi$, $\partial \psi$ is the (regular) subdifferential of $\psi$, and $\text{relint}(C)$ means the relative interior of the set $C$.

We start from introducing the notion of partial smoothness.

**Definition 1.** *A function $\psi$ is partly smooth at a point $W^*$ relative to a set $\mathcal{M}_{W^*} \ni W^*$ if*
1. *Around $W^*$, $\mathcal{M}_{W^*}$ is a $\mathcal{C}^2$-manifold and $\psi|_{\mathcal{M}_{W^*}}$ is $\mathcal{C}^2$.*
2. *$\psi$ is regular (finite with the Fréchet subdifferential coincides with the limiting Fréchet subdifferential) at all points $W \in \mathcal{M}_{W^*}$ around $W^*$ with $\partial\psi(W) \neq \emptyset$.*
3. *The affine span of $\partial\psi(W^*)$ is a translate of the normal space to $\mathcal{M}_{W^*}$ at $W^*$.*
4. *$\partial\psi$ is continuous at $W^*$ relative to $\mathcal{M}_{W^*}$.*

We often call $\mathcal{M}_{W^*}$ the active manifold at $W^*$. Another concept required for manifold identification is prox-regularity (Poliquin & Rockafellar, 1996).

**Definition 2.** *A function $\psi$ is prox-regular at $W^*$ for $V^* \in \partial\psi(W^*)$ if $\psi$ is finite at $W^*$, locally lower semi-continuous around $W^*$, and there is $\rho > 0$ such that $\psi(W_1) \geq \psi(W_2) + \langle V, W_1 - W_2 \rangle - \frac{\rho}{2}\|W_1 - W_2\|^2$ whenever $W_1, W_2$ are close to $W^*$ with $\psi(W_2)$ near $\psi(W^*)$ and $V \in \partial\psi(W_2)$ near $V^*$. $\psi$ is prox-regular at $W^*$ if it is so for all $V \in \partial\psi(W^*)$.*

To broaden the applicable range, a function $\psi$ prox-regular at some $W^*$ is often also assumed to be subdifferentially continuous (Poliquin & Rockafellar, 1996) there, meaning that if $W^t \to W^*$, $\psi(W^t) \to \psi(W^*)$ holds when there are $V^* \in \partial\psi(W^*)$ and a sequence $\{V^t\}$ such that $V^t \in \partial\psi(W^t)$ and $V^t \to V^*$. Notably, all convex and weakly-convex (Nurminskii, 1973) functions are regular, prox-regular, and subdifferentially continuous in their domain.

### 3.1 THEORETICAL RESULTS

When the problem is convex, convergence guarantees for Algorithm 1 under two specific specific schemes are known. First, when $c_t \equiv 1$, RMDA reduces to the classical RDA, and convergence to a global optimum (of $W^t = \tilde{W}^t$ in this case) on convex problems has been proven by Lee & Wright (2012); Duchi & Ruan (2021), with convergence rates of the expected objective or the regret given by Xiao (2010); Lee & Wright (2012). Second, when $c_t = s_{t+1}\alpha_{t+1}^{-1}$ and $(\beta_t, \alpha_t)$ in Line 5 of Algorithm 1 are replaced by $(\beta_{t+1}, \alpha_{t+1})$, convergence is recently analyzed by Kungurtsev & Shikhman (2021). In our analysis below, we do not assume convexity of either term.

We show that if $\{\tilde{W}^t\}$ converges to a point $W^*$ (which could be a non-stationary one), $\{W^t\}$ also converges to $W^*$.

**Lemma 1.** *Consider Algorithm 1 with $\{c_t\}$ satisfying $\sum c_t = \infty$. If $\{\tilde{W}^t\}$ converges to a point $W^*$, $\{W^t\}$ also converges to $W^*$.*

We then show that if $\{\tilde{W}^t\}$ converges to a point, almost surely this point of convergence is stationary. This requires the following lemma for variance reduction of RMDA, meaning that the variance of using $V_t$ to estimate $\nabla f(W^{t-1})$ reduces to zero, as $\alpha_t^{-1}V_t$ converges to $\nabla f(W^{t-1})$ almost surely, and this result could be of its own interest. The first claim below uses a classical result in stochastic optimization that can be found at, for example, (Gupal, 1979, Theorem 4.1, Chapter 2.4), but the second one is, to our knowledge, new.

**Lemma 2.** *Consider Algorithm 1. Assume for any $\xi \sim \mathcal{D}$, $f_\xi$ is $L$-Lipschitz-continuously-differentiable almost surely for some $L$, so $f$ is also $L$-Lipschitz-continuously-differentiable, and there is $C \geq 0$ such that $\mathbb{E}_{\xi_t \sim \mathcal{D}}\|\nabla f_{\xi_t}(W^{t-1})\|^2 \leq C$ for all $t$. If $\{\eta_t\}$ satisfies*

$$\sum \beta_t \eta_t \alpha_t^{-1} = \infty, \quad \sum \left(\beta_t \eta_t \alpha_t^{-1}\right)^2 < \infty, \quad \|W^{t+1} - W^t\|\left(\beta_t \eta_t \alpha_t^{-1}\right)^{-1} \xrightarrow{a.s.} 0, \quad (4)$$

*then $\alpha_t^{-1}V^t \longrightarrow \nabla f(W^{t-1})$ with probability one. Moreover, if $\{W^t\}$ lies in a bounded set, we get $\mathbb{E}\|\alpha_t^{-1}V^t - \nabla f(W^{t-1})\|^2 \to 0$ even if the second condition in (4) is replaced by a weaker condition of $\beta_t \eta_t \alpha_t^{-1} \to 0$.*

In general, the last condition in (4) requires some regularity conditions in $F$ to control the change speed of $W^t$. One possibility is when $\psi$ is the indicator function of a convex set, $\beta_t \eta_t \propto t^p$ for $t \in (1/2, 1)$ will satisfy this condition. However, in other settings for $\eta_t$, even when $F$ and $\psi$ are both convex, existing analyses for the classical RDA such that $c_t \equiv 1$ in Algorithm 1 still need an additional local error bound assumption to control the change of $W^{t+1} - W^t$. Hence, to stay focused on our main message, we take this assumption for granted, and leave finding suitable sufficient conditions for it as future work.

With the help of Lemmas 1 and 2, we can now show the stationarity result for the limit point of the iterates. The assumption of $\beta_t \alpha_t^{-1}$ approaching 0 below is classical in analyses of dual averaging in order to gradually remove the influence of the term $\|W - W^0\|^2$.

**Theorem 1.** *Consider Algorithm 1 with the conditions in Lemmas 1 and 2 hold, and assume the set of stationary points $\mathcal{Z} := \{W \mid 0 \in \partial F(W)\}$ is nonempty and $\beta_t \alpha_t^{-1} \to 0$. For any given $W^0$, consider the event that $\{\tilde{W}^t\}$ converges to a point $W^*$ (each event corresponds to a different $W^*$), then if $\partial\psi$ is outer semicontinuous at $W^*$, and this event has a nonzero probability, $W^* \in \mathcal{Z}$, or equivalently, $W^*$ is a stationary point, with probability one conditional on this event.*

Finally, with Lemmas 1 and 2 and Theorem 1, we prove the main result that the active manifold of the limit point is identified in finite iterations of RMDA under nondegeneracy.

**Theorem 2.** *Consider Algorithm* 1 *with the conditions in Theorem* 1 *satisfied. Consider the event of* $\{\tilde{W}^t\}$ *converging to a certain point* $W^*$ *as in Theorem* 1*, if the probability of this event is nonzero;* $\psi$ *is prox-regular and subdifferentially continuous at* $W^*$ *and partly smooth at* $W^*$ *relative to the active* $\mathcal{C}^2$ *manifold* $\mathcal{M}$; $\partial\psi$ *is outer semicontinuous at* $W^*$; *and the nondegeneracy condition*

$$-\nabla f\left(W^*\right) \in \operatorname{relint} \partial\psi\left(W^*\right) \tag{5}$$

*holds at* $W^*$, *then conditional on this event, almost surely there is* $T_0 \geq 0$ *such that*

$$\tilde{W}^t \in \mathcal{M}, \quad \forall t \geq T_0. \tag{6}$$

*In other words, the active manifold at* $W^*$ *is identified by the iterates of Algorithm* 1 *after a finite number of iterations almost surely.*

As mentioned in Section 1, an important reason for studying manifold identification is to get the lowest-dimensional manifold representing the structure of the limit point, which often corresponds to a preferred property for the application, like the highest sparsity, lowest rank, or lowest VC dimension locally. See an illustrated example in Appendix B.1.

## 4 Applications in Deep Learning

We discuss two popular schemes of training structured deep learning models achieved through regularization to demonstrate the applications of RMDA. More technical details for applying our theory to the regularizers in these applications are in Appendix B.

### 4.1 Structured Sparsity

As modern deep NN models are often gigantic, it is sometimes desirable to trim the model to a smaller one when only limited resources are available. In this case, zeroing out redundant parameters during training at the group level is shown to be useful (Zhou et al., 2016), and one can utilize regularizers promoting structured sparsity for this purpose. The most famous regularizer of this kind is the group-LASSO norm (Yuan & Lin, 2006; Friedman et al., 2010). Given $\lambda \geq 0$ and a collection $\mathcal{G}$ of index sets $\{\mathcal{I}_g\}$ of the variable $W$, this convex regularizer is defined as

$$\psi(W) \coloneqq \lambda \sum_{g=1}^{|\mathcal{G}|} w_g \|W_{\mathcal{I}_g}\|, \tag{7}$$

with $w_g > 0$ being the pre-specified weight for $\mathcal{I}_g$. For any $W^*$, let $\mathcal{G}_{W^*} \subseteq \mathcal{G}$ be the index set such that $W^*_{\mathcal{I}_j} = 0$ for all $j \in \mathcal{G}_{W^*}$, the group-LASSO norm is partly smooth around $W^*$ relative to the manifold $\mathcal{M}_{W^*} \coloneqq \{W \mid W_{\mathcal{I}_i} = 0, \forall i \in \mathcal{G}_{W^*}\}$, so our theory applies.

In order to promote structured sparsity, we need to carefully design the grouping. Fortunately, in NNs, the parameters can be grouped naturally (Wen et al., 2016). For any fully-connected layer, let $W \in \mathbb{R}^{\text{out}\times\text{in}}$ be the matrix representation of the associated parameters, where out is the number of output neurons and in is that of input neurons, we can consider the column-wise groups, defined as $W_{:,j}$ for all $j$, and the row-wise groups of the form $W_{i,:}$. For a convolutional layer with $W \in \mathbb{R}^{\text{filter}\times\text{channel}\times\text{height}\times\text{width}}$ being the tensor form of the corresponding parameters, we can consider channel-wise, filter-wise, and kernel-wise groups, defined respectively as $W_{:,j,:,:}$, $W_{i,:,:,:}$ and $W_{i,j,:,:}$.

### 4.2 Binary/Discrete Neural Networks

Making the parameters of an NN binary integers is another way to obtain a more compact model during training and deployment (Hubara et al., 2016), but discrete optimization is hard to scale-up. Using a vector representation $w \in \mathbb{R}^m$ of the variables, Hou et al. (2017) thus proposed to use the indicator function of $\{w \mid w_{\mathcal{I}_i} = \alpha_i b_{\mathcal{I}_i}, \alpha_i > 0, b_{\mathcal{I}_i} \in \{\pm 1\}^{|\mathcal{I}_i|}\}$ to induce the entries of $w$ to be binary without resorting to discrete optimization tools, where each $\mathcal{I}_i$ enumerates all parameters in the $i$-th layer. Yang et al. (2019) later proposed to use $\min_{\alpha \in [0,1]^m} \quad \sum_{i=1}^{m} \left(\alpha_i(w_i+1)^2 + (1-\alpha_i)(w_i-1)^2\right)$ as the regularizer and to include $\alpha$ as

a variable to train. At any $\alpha^*$ with $I_0 \coloneqq \{i \mid \alpha_i^* = 0\}$ and $I_1 \coloneqq \{i \mid \alpha_i^* = 1\}$, the objective is partly smooth relative to the manifold $\{(W, \alpha) \mid \alpha_{I_0} = 0, \alpha_{I_1} = 1\}$. Extension to discrete NNs beyond the binary ones is possible, and Bai et al. (2019) have proposed regularizers with closed-form proximal operators for it.

## 5 Experiments

We use the structured sparsity application in Section 4.1 to empirically exemplify the ability of RMDA to find desired structures in the trained NNs. RMDA and the following methods for structured sparsity in deep learning are compared using PyTorch (Paszke et al., 2019).

- ProxSGD (Yang et al., 2019): A simple proxMSGD algorithm. To obtain group sparsity, we skip the interpolating step in Yang et al. (2019).
- ProxSSI (Deleu & Bengio, 2021): This is a special case of the adaptive proximal SGD framework of Yun et al. (2021) that uses the Newton-Raphson algorithm to approximately solve the subproblem. We directly use the package released by the authors.

We exclude the algorithm of Wen et al. (2016) because their method is shown to be worse than ProxSSI by Deleu & Bengio (2021).

To compare these algorithms, we examine both the validation accuracy and the group sparsity level of their trained models. We compute the group sparsity as the percentage of groups whose elements are all zero, so the reported group sparsity is zero when there is no group with a zero norm, and is one when the whole model is zero. For all methods above, we use (7) with column-wise and channel-wise groupings in the regularization for training, but adopt the kernel-wise grouping in their group sparsity evaluation. Throughout the experiments, we always use multi-step learning rate scheduling that decays the learning rate by a constant factor every time the epoch count reaches a pre-specified threshold. For all methods, we conduct grid searches to find the best hyperparameters. All results shown in tables in Sections 5.1 and 5.2 are the mean and standard deviation of three independent runs with the same hyperparameters, while figures use one representative run for better visualization.

In convex optimization, a popular way to improve the practical convergence behavior for momentum-based methods is restarting that periodically reset the momentum to zero (O'donoghue & Candes, 2015). Following this idea, we introduce a restart heuristic to RMDA. At each round, we use the output of Algorithm 1 from the previous round as the new input to the same algorithm, and continue using the scheduling $\eta$ and $c$ without resetting them. For $\psi \equiv 0$, Jelassi & Defazio (2020) suggested to increase $c_t$ proportional to the decrease of $\eta_t$ until reaching $c_t = 1$. We adopt the same setting for $c_t$ and $\eta_t$ and restart RMDA whenever $\eta_t$ changes. As shown in Section 3 that $\tilde{W}^t$ finds the active manifold, increasing $c_t$ to 1 also accords with our interest in identifying the stationary structure.

### 5.1 Correctness of Identified Structure Using Synthetic Data

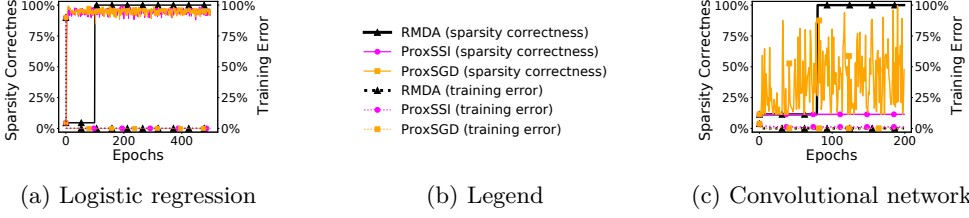

(a) Logistic regression      (b) Legend      (c) Convolutional network

Figure 1: Group sparsity pattern correctness and training error rates on synthetic data.

Our first step is to numerically verify that RMDA can indeed identify the stationary structure desired. To exactly find a stationary point and its structure a priori, we consider synthetic problems. We first decide a ground truth model $W$ that is structured sparse, generate random data points that can be well separated by $W$, and then decide their labels using $W$. The generated data are then taken as our training data. We consider a linear logistic

regression model and a small NN that has one fully-connected layer and one convolutional layer. To ensure convergence to the ground truth, for logistic regression we generate more data points than the problem dimension to ensure the problem is strongly convex so that there is only one stationary/optimal point, and for the small NN, we initialize all algorithms close enough to the ground truth. We report in Fig. 1 training error rates (as an indicator for the proximity to the ground truth) and percentages of the optimal group sparsity pattern of the ground truth identified. Clearly, although all methods converge to the ground truth, only RMDA identifies the correct structure of it, and other methods without guarantees for manifold identification fail.

## 5.2 Neural Networks with Real Data

Table 1: Group sparsity and validation accuracy of different methods. We report mean and standard deviation of three independent runs (except that for the linear convex model, only one run is conducted as we are guaranteed to find the global optima). MSGD is the baseline with no sparsity-inducing regularizer.

| Algorithm | Validation accuracy | Group sparsity | Validation accuracy | Group sparsity |
|---|---|---|---|---|
| | Logistic regression/MNIST | | Fully-connected NN/FashionMNIST | |
| ProxSGD | 91.31 % | 38.78 % | $88.72 \pm 0.05\%$ | $31.42 \pm 0.36\%$ |
| ProxSSI | 91.31 % | 39.54 % | $88.44 \pm 0.41\%$ | $35.25 \pm 1.56\%$ |
| RMDA | **91.34 %** | **56.51 %** | **$88.09 \pm 0.04\%$** | **$42.89 \pm 0.66\%$** |
| | LeNet5/MNIST | | LeNet5/FashionMNIST | |
| MSGD | $99.36 \pm 0.06\%$ | - | $91.96 \pm 0.01\%$ | - |
| ProxSGD | **$99.13 \pm 0.02\%$** | $76.57 \pm 2.33\%$ | $90.99 \pm 0.17\%$ | $50.50 \pm 2.66\%$ |
| ProxSSI | $99.07 \pm 0.03\%$ | $77.82 \pm 1.56\%$ | $90.93 \pm 0.02\%$ | $60.49 \pm 1.05\%$ |
| RMDA | $99.10 \pm 0.06\%$ | **$79.81 \pm 1.56\%$** | **$91.41 \pm 0.10\%$** | **$66.15 \pm 1.68\%$** |
| | VGG19/CIFAR10 | | VGG19/CIFAR100 | |
| MSGD | $94.03 \pm 0.11\%$ | - | $74.62 \pm 0.22\%$ | - |
| ProxSGD | $92.38 \pm 0.31\%$ | $72.57 \pm 6.04\%$ | $71.91 \pm 0.08\%$ | $08.63 \pm 4.88\%$ |
| ProxSSI | $92.51 \pm 0.03\%$ | $81.05 \pm 0.16\%$ | $66.20 \pm 0.38\%$ | $46.41 \pm 1.42\%$ |
| RMDA | **$93.62 \pm 0.15\%$** | **$86.37 \pm 0.25\%$** | **$72.23 \pm 0.20\%$** | **$58.86 \pm 0.41\%$** |
| | ResNet50/CIFAR10 | | ResNet50/CIFAR100 | |
| MSGD | $95.65 \pm 0.03\%$ | - | $79.13 \pm 0.19\%$ | - |
| ProxSGD | $92.36 \pm 0.14\%$ | $76.82 \pm 4.09\%$ | $75.53 \pm 0.49\%$ | $51.83 \pm 0.34\%$ |
| ProxSSI | $94.09 \pm 0.08\%$ | $74.81 \pm 1.28\%$ | $74.52 \pm 0.29\%$ | $32.79 \pm 2.53\%$ |
| RMDA | **$94.25 \pm 0.02\%$** | **$83.01 \pm 0.50\%$** | **$76.12 \pm 0.46\%$** | **$57.67 \pm 3.76\%$** |

We turn to real-world data used in modern computer vision problems. We consider two rather simple models and six more complicated modern CNN cases. The two simpler models are linear logistic regression with the MNIST dataset (LeCun et al., 1998), and training a small NN with seven fully-connected layers on the FashionMNIST dataset (Xiao et al., 2017). The six more complicated cases are:

1. A version of LeNet5 with the MNIST dataset,
2. The same version of LeNet5 with the FashionMNIST dataset,
3. A modified VGG19 (Simonyan & Zisserman, 2015) with the CIFAR10 dataset (Krizhevsky, 2009),
4. The same modified VGG19 with the CIFAR100 dataset (Krizhevsky, 2009),
5. ResNet50 (He et al., 2016) with the CIFAR10 dataset, and
6. ResNet50 with the CIFAR100 dataset.

For these six more complicated tasks, we include a dense baseline of MSGD with no sparsity-inducing regularizer in our comparison. For all training algorithms on VGG19 and ResNet50, we follow the standard practice in modern vision tasks to apply data augmentation through random cropping and horizontal flipping so that the training problem is no longer a finite-sum one. From Fig. 2, we see that similar to the previous experiment, the group sparsity level of RMDA is stable in the last epochs, while that of ProxSGD and ProxSSI oscillates below. This suggests that RMDA is the only method that, as proven in Section 3, identifies the structured sparsity at its limit point, and other methods with no variance reduction fail. Moreover, Table 1 shows that manifold identification of RMDA is achieved with no

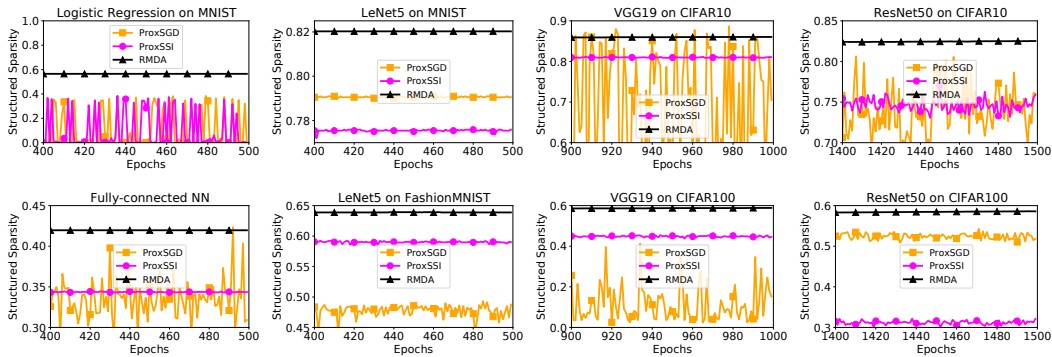

Figure 2: Group Sparsity v.s epochs of different algorithms on NNs of a single run.

sacrifice of the validation accuracy, so RMDA beats ProxSGD and ProxSSI in both criteria, and its accuracy is close to that of the dense baseline of MSGD. Moreover, for VGG19 and ResNet50, RMDA succeeds in finding the optimal structured sparsity pattern despite the presence of data augmentation, showing that RMDA can indeed overcome the difficulty from the infinite-sum setting of modern deep learning tasks.

We also report that in the ResNet50/CIFAR100 task, on our NVIDIA RTX 8000 GPU, MSGD, ProxSGD, and RMDA have similar per-epoch cost of 68, 77, and 91 seconds respectively, while ProxSSI needs 674 seconds per epoch. RMDA is thus also more suitable for large-scale structured deep learning in terms of practical efficiency.

## 5.3 Comparison with Pruning

We compare RMDA with a state-of-the-art pruning method RigL (Evci et al., 2020). As pruning focuses on unstructured sparsity, we use RMDA with $\psi(W) = \lambda \|W\|_1$ to have a fair comparison, and tune $\lambda$ to achieve a pre-specified sparsity level. We run RigL with 1000 epochs, as its performance at the default 500 epochs was unstable, and let RMDA use the same number of epochs. Results of 98% sparsity in Table 2 show that RMDA consistently outdoes RigL, indicating regularized training could be a promising alternative to pruning.

Table 2: Comparison between RMDA and RigL with 1000 epochs for unstructured sparsity in a single run.

| Algorithm | ResNet50 with CIFAR10 | | ResNet50 with CIFAR100 | |
|---|---|---|---|---|
| | Sparsity | Accuracy | Sparsity | Accuracy |
| Dense baseline | | 94.81% | | 74.61% |
| RMDA | **98.36%** | **93.78%** | **98.32%** | **74.32%** |
| RigL | 98.00% | 93.41% | 98.00% | 70.88% |

## 6 Conclusions

In this work, we proposed and analyzed a new algorithm, RMDA, for efficiently training structured neural networks with state-of-the-art performance. Even in the presence of data augmentation, RMDA can still achieve variance reduction and provably identify the desired structure at a stationary point using the tools of manifold identification. Experiments show that existing algorithms for the same purpose fail to find a stable stationary structure, while RMDA achieves so with no accuracy drop nor additional time cost.

## Acknowledgements

This work was supported in part by MOST of R.O.C. grant 109-2222-E-001-003-MY3, and the AWS Cloud Credits for Research program of Amazon Inc.

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

# Appendices

## Table of Contents

## A    PROOFS

### A.1    PROOF OF LEMMA 1

*Proof.* Using (3), the distance between $W^t$ and $W^*$ can be upper bounded through the triangle inequality:

$$\left\| W^t - W^* \right\| = \left\| (1 - c_t) \left( W^{t-1} - W^* \right) + c_t \left( \tilde{W}^t - W^* \right) \right\|$$
$$\leq c_t \left\| \tilde{W}^t - W^* \right\| + (1 - c_t) \left\| W^{t-1} - W^* \right\|. \tag{8}$$

For any event such that $\tilde{W}^t \to W^*$, for any $\epsilon > 0$, we can find $T_\epsilon \geq 0$ such that

$$\left\| \tilde{W}^t - W^* \right\| \leq \epsilon, \quad \forall t \geq T_\epsilon.$$

Let $\delta_t := \|W^t - W^*\|$, we see from the above inequality and (8) that

$$\delta_t \leq (1 - c_t)\, \delta_{t-1} + c_t \epsilon, \quad \forall t \geq T_\epsilon.$$

By deducting $\epsilon$ from both sides, we get that

$$(\delta_t - \epsilon) \leq (1 - c_t)(\delta_{t-1} - \epsilon), \quad \forall t \geq T_\epsilon.$$

Since $\sum c_t = \infty$, we further deduce that

$$\lim_{t \to \infty} (\delta_t - \epsilon) \leq \prod_{t=T_\epsilon}^{\infty} (1 - c_t)(\delta_{T_\epsilon - 1} - \epsilon)$$

$$\leq \prod_{t=T_\epsilon}^{\infty} \exp(-c_t)(\delta_{T_\epsilon - 1} - \epsilon)$$

$$= \exp\left( -\sum_{t=T_\epsilon}^{\infty} c_t \right)(\delta_{T_\epsilon - 1} - \epsilon) = 0,$$

where in the first inequality we used the fact that $\exp(x) \geq 1 + x$ for all real number $x$. The result above then implies that

$$\lim_{t \to \infty} \delta_t \leq \epsilon.$$

As $\epsilon$ is arbitrary and $\delta_t \geq 0$ from the definition, we conclude that $\lim_{t \to \infty} \delta_t = 0$, which is equivalent to that $W^t \to W^*$. $\qquad \square$

### A.2 Proof of Lemma 2

*Proof.* We observe that

$$\alpha_t^{-1} V^t = \sum_{k=1}^{t} \frac{\eta_k \beta_k}{\alpha_t} \nabla f_{\xi_k}\left( W^{k-1} \right)$$

$$= \frac{\alpha_{t-1}}{\alpha_t} \alpha_{t-1}^{-1} V^{t-1} + \frac{\alpha_t - \alpha_{t-1}}{\alpha_t} \nabla f_{\xi_t}\left( W^{t-1} \right)$$

$$= \left( 1 - \frac{\beta_t \eta_t}{\alpha_t} \right) \alpha_{t-1}^{-1} V^{t-1} + \frac{\beta_t \eta_t}{\alpha_t} \nabla f_{\xi_t}\left( W^{t-1} \right).$$

From that $f$ is $L$-Lipschitz-continuously differentiable, we have that

$$\left\| \mathbb{E}_{\xi_{t+1} \sim \mathcal{D}} \left[ \nabla f_{\xi_{t+1}}\left( W^t \right) \right] - \mathbb{E}_{\xi_t \sim \mathcal{D}} \left[ \nabla f_{\xi_t}\left( W^{t-1} \right) \right] \right\| = \left\| \nabla f\left( W^t \right) - f\left( W^{t-1} \right) \right\|$$
$$\leq L \left\| W^t - W^{t-1} \right\|. \tag{9}$$

Therefore, (4) and (9) imply that

$$0 \leq \frac{\left\| \mathbb{E}_{\xi_{t+1} \sim \mathcal{D}} \left[ \nabla f_{\xi_{t+1}}\left( W^t \right) \right] - \mathbb{E}_{\xi_t \sim \mathcal{D}} \left[ \nabla f_{\xi_t}\left( W^{t-1} \right) \right] \right\|}{\beta_t \eta_t \alpha_t^{-1}} \leq L \frac{\left\| W^t - W^{t-1} \right\|}{\beta_t \eta_t \alpha_t^{-1}} \xrightarrow{\text{a.s.}} 0,$$

which together with the sandwich lemma shows that

$$\frac{\left\| \mathbb{E}_{\xi_{t+1} \sim \mathcal{D}} \left[ \nabla f_{\xi_{t+1}}\left( W^t \right) \right] - \mathbb{E}_{\xi_t \sim \mathcal{D}} \left[ \nabla f_{\xi_t}\left( W^{t-1} \right) \right] \right\|}{\beta_t \eta_t \alpha_t^{-1}} \xrightarrow{\text{a.s.}} 0. \tag{10}$$

Therefore, the first two conditions of (4) together with (10) and the bounded variance assumption satisfy the requirements of (Gupal, 1979, Chapter 2.4, Theorem 4.1), so the conclusion of almost sure convergence hold.

For the convergence in $\mathcal{L}_2$ part, we first define $m^t := \alpha_t^{-1} V_t$ and $\tau_t := \beta_t \eta_t \alpha_t^{-1}$ for notational ease. Consider $\left\| m^{t+1} - \nabla F(W^t) \right\|^2$, we have from the update rule in Algorithm 1 that

$$
\begin{aligned}
&\left\| m^{t+1} - \nabla F(W^t) \right\|^2 \\
&= \left\| (1-\tau_t) m^t + \tau_t \nabla f_{\xi_{t+1}}(W^t) - \nabla F(W^t) \right\|^2 \\
&= \left\| (1-\tau_t)\left( m^t - \nabla F(W^t) \right) + \tau_t \left( \nabla f_{\xi_{t+1}}(W^t) - \nabla F(W^t) \right) \right\|^2 \\
&= (1-\tau_t)^2 \left\| m^t - \nabla F(W^t) \right\|^2 + \tau_t^2 \left\| \nabla f_{\xi_{t+1}}(W^t) - \nabla F(W^t) \right\|^2 \\
&\qquad + 2\tau_t(1-\tau_t)\langle m^t - \nabla F(W^t), \nabla f_{\xi_{t+1}}(W^t) - \nabla F(W^t)\rangle \\
&= (1-\tau_t)^2 \left\| \left( m^t - \nabla F\left(W^{t-1}\right)\right) + \left(\nabla F\left(W^{t-1}\right) - \nabla F\left(W^t\right)\right) \right\|^2 \qquad (11) \\
&\qquad + \tau_t^2 \left\| \nabla f_{\xi_{t+1}}(W^t) - \nabla F(W^t) \right\|^2 + 2\tau_t(1-\tau_t)\langle m^t - \nabla F(W^t), \nabla f_{\xi_{t+1}}(W^t) - \nabla F(W^t)\rangle.
\end{aligned}
$$

Let $\{\mathcal{F}_t\}_{t\geq 0}$ denote the natural filtration of $\{(m^t, W^t)\}_{t\geq 0}$. Namely, $\mathcal{F}_t$ records the information of $W^0$, $\{c_i\}_{i=0}^{t-1}$, $\{\eta_i\}_{i=0}^{t-1}$, and $\{\xi_i\}_{i=1}^{t}$. By defining $U_t := \left\| m^t - \nabla F(W^{t-1}) \right\|^2$ and taking expectation over (11) conditional on $\mathcal{F}_t$, we obtain from $\mathbb{E}\left[\nabla f_{\xi_{t+1}}(W^t) \mid \mathcal{F}_t\right] = \nabla F(W^t)$ that

$$
\begin{aligned}
\mathbb{E}\left[U_{t+1} \mid \mathcal{F}_t\right] &= (1-\tau_t)^2 \left\| \left(m^t - \nabla F\left(W^{t-1}\right)\right) + \left(\nabla F\left(W^{t-1}\right) - \nabla F\left(W^t\right)\right) \right\|^2 \\
&\qquad + \tau_t^2 \mathbb{E}\left[\left\| \nabla f_{\xi_t}(W^t) - \nabla F(W^t) \right\|^2 \mid \mathcal{F}_t\right]. \qquad (12)
\end{aligned}
$$

From the last condition in (4) and the Lipschitz continuity of $\nabla F$, there are random variables $\{\epsilon_t\}$ and $\{u_t\}$ such that $\|u_t\| = 1$, $\epsilon_t \geq 0$, and $\nabla F(W^{t-1}) - \nabla F(W^t) = \tau_t \epsilon_t u_t$ for all $t > 0$, with $\epsilon_t \downarrow 0$ almost surely. We thus obtain that

$$
\begin{aligned}
&\left\| m^t - \nabla F(W^{t-1}) + \nabla F(W^{t-1}) - \nabla F(W^t) \right\|^2 \\
&= \left\| m^t - \nabla F(W^{t-1}) + \tau_t \epsilon_t u_t \right\|^2 \\
&= (1+\tau_t)^2 \left\| \frac{1}{1+\tau_t}\left(m^t - \nabla F\left(W^{t-1}\right)\right) + \frac{\tau_t}{1+\tau_t}\epsilon_t u_t \right\|^2 \\
&\leq (1+\tau_t)^2 \left( \frac{1}{1+\tau_t} U_t + \frac{\tau_t}{1+\tau_t}\epsilon_t^2 \right), \qquad (13)
\end{aligned}
$$

where we used Jensen's inequality and the convexity of $\|\cdot\|^2$ in the last inequality. By substituting (13) back into (12), we obtain

$$
\begin{aligned}
&\mathbb{E}\left[U_{t+1} | \mathcal{F}_t\right] \\
&\leq (1-\tau_t)^2(1+\tau_t) U_t + (1-\tau_t)^2(1+\tau_t)\tau_t \epsilon_t^2 + \tau_t^2 \mathbb{E}\left[\left\| \nabla f_{\xi_t}(W^t) - \nabla F(W^t) \right\|^2 \mid \mathcal{F}_t\right] \\
&\leq (1-\tau_t)(U_t + \tau_t \epsilon_t^2) + \tau_t^2 \mathbb{E}\left[\left\| \nabla f_{\xi_t}(W^t) - \nabla F(W^t) \right\|^2 \mid \mathcal{F}_t\right] \\
&\leq (1-\tau_t) U_t + \tau_t \epsilon_t^2 + \tau_t^2 \mathbb{E}\left[\left\| \nabla f_{\xi_t}(W^t) - \nabla F(W^t) \right\|^2 \mid \mathcal{F}_t\right]. \qquad (14)
\end{aligned}
$$

For the last term in (14), we notice that

$$
\begin{aligned}
\mathbb{E}\left[\left\| \nabla f_{\xi_t}(W^t) - \nabla F(W^t) \right\|^2 \mid \mathcal{F}_t\right] &\leq 2\left( \mathbb{E}\left[\left\| \nabla f_{\xi_t}(W^t) \right\|^2\right] + \left\| \nabla F(W^t) \right\|^2 \right) \\
&\leq 2\left( C + \left\| \nabla F(W^t) \right\|^2 \right), \qquad (15)
\end{aligned}
$$

where the last inequality is from the bounded variance assumption. Since by assumption the $\{W^t\}$ lies in a bounded set $K$, we have that for any point $W^* \in K$, $W^t - W^*$ is upper bounded, and thus $\left\| \nabla F(W^t) - \nabla F(W^*) \right\|$ is also bounded, implying that $\left\| \nabla F(W^t) \right\|^2 \leq C_2$ for some $C_2 \geq 0$. Therefore, (15) further leads to

$$
\mathbb{E}\left[\left\| \nabla f_{\xi_t}(W^t) - \nabla F(W^t) \right\|^2 \mid \mathcal{F}_t\right] \leq C_3 \qquad (16)
$$

for some $C_3 \geq 0$.

Now we further take expectation on (14) and apply (16) to obtain

$$\mathbb{E}U_{t+1} \leq (1 - \tau_t)\mathbb{E}U_t + \tau_t\epsilon_t^2 + \tau_t^2 C_3 = (1 - \tau_t)\mathbb{E}U_t + \tau_t\left(\epsilon_t^2 + \tau_t C_3\right). \qquad (17)$$

Note that the third implies $\epsilon_t \downarrow 0$, so this together with the second condition that $\tau_t \downarrow 0$ means $\epsilon_t^2 + \tau_t C_3 \downarrow 0$ as well, and thus for any $\delta > 0$, we can find $T_\delta \geq 0$ such that $\epsilon_t^2 + \tau_t C_3 \leq \delta$ for all $t \geq T_\delta$. Thus, (17) further leads to

$$\mathbb{E}U_{t+1} - \delta \leq (1 - \tau_t)\mathbb{E}U_t + \tau_t\delta - \delta = (1 - \tau_t)\left(\mathbb{E}U_t - \delta\right), \forall t \geq T_\delta. \qquad (18)$$

This implies that $(\mathbb{E}U_t - \delta)$ becomes a decreasing sequence starting from $t \geq T_\delta$, and since $U_t \geq 0$, this sequence is lower bounded by $-\delta$, and hence it converges to a certain value. By recursion of (18), we have that

$$\mathbb{E}U_t - \delta \leq \prod_{i=T_\delta}^{t}(1 - \tau_i)\left(\mathbb{E}U_{T_\delta} - \delta\right),$$

and from the well-known inequality $(1 + x) \leq \exp^x$ for all $x \in \mathcal{R}$, the above result leads to

$$\mathbb{E}U_t - \delta \leq \exp\left(-\sum i = T_\delta{}^t\tau_i\right)\left(\mathbb{E}U_{T_\delta} - \delta\right).$$

By letting $t$ approach infinity and noting that the first condition of (4) indicates

$$\sum_{t=k}^{\infty}\tau_t = \infty$$

for any $k \geq 0$, we see that

$$-\delta \leq \lim_{t \to \infty}\mathbb{E}U_t - \delta \leq \exp\left(-\sum_{i=T_\delta}^{\infty}\tau_i\right)\left(\mathbb{E}U_{T_\delta} - \delta\right) = 0. \qquad (19)$$

As $\delta$ is arbitrary, by taking $\delta \downarrow 0$ in (19) and noting the nonnegativity of $U_t$, we conclude that $\lim \mathbb{E}U_t = 0$, as desired. This proves the last result in Lemma 2. $\qquad \square$

### A.3   PROOF OF THEOREM 1

*Proof.* Using Lemma 2, we can view $\alpha_t^{-1}V^t$ as $\nabla f(W^t)$ plus some noise that asymptotically decreases to zero with probability one:

$$\alpha_t^{-1}V_t = \nabla f(W^t) + \epsilon_t, \quad \|\epsilon_t\| \xrightarrow{\text{a.s.}} 0. \qquad (20)$$

We use (20) to rewrite the optimality condition of (2) as (also see Line 5 of Algorithm 1)

$$-\left(\nabla f\left(W^t\right) + \epsilon_t + \beta_t\alpha_t^{-1}\left(\tilde{W}^t - W^0\right)\right) \in \partial\psi\left(\tilde{W}^t\right). \qquad (21)$$

Now we consider $\partial F(\tilde{W}^t)$. Clearly from (21), we have that

$$\nabla f\left(\tilde{W}^t\right) - \nabla f\left(W^t\right) - \epsilon_t - \beta_t\alpha_t^{-1}\left(\tilde{W}^t - W^0\right) \in \partial\nabla f\left(\tilde{W}^t\right) + \psi\left(\tilde{W}^t\right) = \partial F\left(\tilde{W}^t\right). \qquad (22)$$

Now we consider the said event that $\tilde{W}^t \to W^*$ for a certain $W^*$, and let us define this event as $\mathcal{A} \subseteq \Omega$. From Lemma 1, we know that $W^t \to W^*$ as well under $\mathcal{A}$. Let us define $\mathcal{B} \subseteq \Omega$ as the event of $\epsilon_t \to 0$, then we know that since $P(\mathcal{A}) > 0$ and $P(\mathcal{B}) = 1$, where $P$ is the probability function for events in $\Omega$, $P(\mathcal{A} \cap \mathcal{B}) = P(\mathcal{A})$. Therefore, conditional on the event of $\mathcal{A}$, we have that $\epsilon_t \xrightarrow{\text{a.s.}} 0$ still holds. Now we consider any realization of $\mathcal{A} \cap \mathcal{B}$. For the right-hand side of (22), as $\tilde{W}^t$ is convergent and $\beta_t\alpha_t^{-1}$ decreases to zero, by letting $t$ approach infinity, we have that

$$\lim_{t \to \infty}\epsilon_t + \beta_t\alpha_t^{-1}\left(\tilde{W}^t - W^0\right) = 0 + 0\left(W^* - W^0\right) = 0.$$

By the Lipschitz continuity of $\nabla f$, we have from (3) and (4) that

$$0 \leq \left\|\nabla f\left(\tilde{W}^t\right) - \nabla f\left(W^t\right)\right\| \leq L\left\|W^t - \tilde{W}^t\right\|.$$

As $\{W^t\}$ and $\{\tilde{W}^t\}$ converge to the same point, we see that $\|W^t - \tilde{W}^t\| \to 0$, so $\nabla f\left(\tilde{W}^t\right) - \nabla f(W^t)$ also approaches zero. Hence, the limit of the right-hand side of (22) is

$$\lim_{t\to\infty} \nabla f\left(\tilde{W}^t\right) - \left(\nabla f\left(W^t\right) + \epsilon_t + \beta_t \alpha_t^{-1}\left(\tilde{W}^t - W^0\right)\right) = 0. \tag{23}$$

On the other hand, for the left-hand side of (22), the outer semicontinuity of $\partial\psi$ at $W^*$ and the continuity of $\nabla f$ show that

$$\lim_{t\to\infty} \nabla f(\tilde{W}^t) + \partial\psi(\tilde{W}^t) \subseteq \partial\nabla f(W^*) + \psi\left(W^*\right) = \partial F(W^*). \tag{24}$$

Substituting (23) and (24) back into (22) then proves that $0 \in \partial F(W^*)$ and thus $W^* \in \mathcal{Z}$. $\qquad\square$

## A.4 Proof of Theorem 2

*Proof.* Our discussion in this proof are all under the event that $\tilde{W}^t \to W^*$. From the argument in Appendix A.3, we can view $\alpha_t^{-1}V^t$ as $\nabla f(W^t)$ plus some noise that asymptotically decreases to zero with probability one as shown in (20). From Lemma 1, we know that $W^t \to W^*$. From (21), there is $U^t \in \partial\psi\left(\tilde{W}^t\right)$ such that

$$U^t = -\alpha_t^{-1}V^t + \alpha_t^{-1}\beta_t\left(\tilde{W}^t - W^0\right). \tag{25}$$

Moreover, we define

$$\gamma_t \coloneqq W^t - \tilde{W}^t. \tag{26}$$

By combining (25)–(26) with (20), we obtain

$$\min_{Y\in\partial F(\tilde{W}^t)} \|Y\|$$
$$\leq \left\|\nabla f\left(\tilde{W}^t\right) + U^t\right\|$$
$$= \left\|\nabla f\left(\tilde{W}^t\right) - \nabla f\left(W^t\right) - \epsilon_t - \alpha_t^{-1}\beta_t\left(\tilde{W}^t - W^0\right)\right\|$$
$$\leq \left\|\nabla f\left(\tilde{W}^t\right) - \nabla f\left(W^t\right)\right\| + \|\epsilon_t\| + \alpha_t^{-1}\beta_t\|\tilde{W}^t - W^0\|$$
$$\leq L\|\gamma_t\| + \|\epsilon_t\| + \alpha_t^{-1}\beta_t\left(\left\|W^* - \tilde{W}^t\right\| + \left\|W^0 - W^*\right\|\right), \tag{27}$$

where we used the Lipschitz continuity of $\nabla f$ and the triangle inequality in the last inequality.

We now separately bound the terms in (27). From that $W^t \to W^*$ and $\tilde{W}^t \to W^*$, it is straightforward that $\|\gamma_t\| \to 0$. The second term decreases to zero almost surely according to (20) and the argument in Appendix A.3. For the last term, since $\alpha_t^{-1}\beta_t \to 0$, and $\left\|\tilde{W}^t - W^*\right\| \to 0$, we know that

$$\alpha_t^{-1}\beta_t\left\|W^0 - W^*\right\| \to 0, \quad \alpha_t^{-1}\beta_t\left\|\tilde{W}^t - W^*\right\| \to 0.$$

Therefore, we conclude from the above argument and (27) that

$$\min_{Y\in\partial F(\tilde{W}^t)} \|Y\| \xrightarrow{\text{a.s.}} 0.$$

As $f$ is smooth with probability one, we know that if $\psi$ is partly smooth at $W^*$ relative to $\mathcal{M}$, then so is $F = f + \psi$ with probability one. Moreover, Lipschitz-continuously differentiable functions are always prox-regular, and the sum of two prox-regular functions is still prox-regular, so $F$ is also prox-regular at $W^*$ with probability one. Following the argument identical to that in Appendix A.3, we know that these probability one events are still probability one conditional on the event of $\tilde{W}^t \to W^*$ as this event has a nonzero probability. As $\tilde{W}^t \to W^*$ and $\nabla f(\tilde{W}^t) + U^t \xrightarrow{\text{a.s.}} 0 \in \partial F(W^*)$ (the inclusion is from (5)), we have from the subdifferential continuity of $\psi$ and the smoothness of $f$ that $F(\tilde{W}^t) \xrightarrow{\text{a.s.}} F(W^*)$. Since we also have $\tilde{W}^t \to W^*$ and $\min_{Y\in\partial F(\tilde{W}^t)} \|Y\| \xrightarrow{\text{a.s.}} 0$, clearly

$$\left(\tilde{W}^t, F\left(W^t\right), \min_{Y\in\partial F(\tilde{W}^t)} \|Y\|\right) \xrightarrow{\text{a.s.}} (W^*, F(W^*), 0). \tag{28}$$

Therefore, (28) and (5) together with the assumptions on $\psi$ at $W^*$ imply that with probability one, all conditions of Lemma 1 of Lee (2020) are satisfied, so from it, (6) holds almost surely, conditional on the event of $\tilde{W}^t \to W^*$. $\qquad\square$

# B  Additional Discussions on Applications

We now discuss in more technical details the applications in Section 4.1, especially regarding how the regularizers satisfy the properties required by our theory.

## B.1  Structured Sparsity

We start our discussion with the simple $\ell_1$ norm as the warm-up for the group-LASSO norm. It is clear that $\|W\|_1$ is a convex function that is finite everywhere, so it is prox-regular, subdifferentially continuous, and regular everywhere, hence we just need to discuss about the remaining parts in Definition 1. Consider a problem with dimension $n > 0$. Note that

$$\|x\|_1 = \sum_{i=1}^{n} |x_i|,$$

and the absolute value is smooth everywhere except the point of origin. Therefore, it is clear that $\|x\|_1$ is locally smooth if $x_i \neq 0$ for all $i$. For any point $x^*$, when there is an index set $I$ such that $x_i^* = 0$ for all $i \in I$ and $x_i^* \neq 0$ for $i \notin I$, we see that the part of the norm corresponds to $I^C$ (the complement of $I$):

$$\sum_{i \in I^C} |x_i^*|$$

is locally smooth around $x^*$. Without loss of generality, we assume that $I = \{1, 2, \ldots, k\}$ for some $k \geq 0$, then the subdifferential of $\|x\|_1$ at $x^*$ is the set

$$\{\operatorname{sgn}(x_1)\} \times \cdots \times \{\operatorname{sgn}(x_k)\} \times [-1, 1]^{n-k}, \tag{29}$$

and clearly if we move from $x^*$ along any direction $y := (y_1, \ldots, y_k, 0, \ldots, 0)$ with a small step, the function value changes smoothly as it is a linear function, satisfying the first condition of Definition 1. Along the same direction $y$ with a small enough step, the set of subdifferential remains the same, so the continuity of subdifferential requirement holds. We can also observe from the above argument that the manifold should be $\mathcal{M}_{x^*} = \{x \mid x_i = 0, \forall i \in I\}$, and clearly it is a subspace of $\mathbb{R}^n$ with its normal space at $x^*$ being $N := \{y \mid \langle x^*, y \rangle = 0\} = \{y \mid y_i = 0, \forall i \in I^C\}$, which is clearly the affine span of (29) with the translation being $(\operatorname{sgn}(x_1) \times \cdots \times \operatorname{sgn}(x_k), 0, \ldots, 0)$. Moreover, indeed the manifolds are low dimensional ones, and for iterates approaching $x^*$, staying in this active manifold means that the (unstructured) sparsity of the iterates is the same as the limit point $x^*$. We also provide a graphical illustration of $\|x\|_1$ with $n = 2$ in Fig. 3. We can observe that for any $x$ with $x_1 \neq 0$ and $x_2 \neq 0$, the function is smooth locally around any point, meaning that $\|x\|_1$ is partly smooth relative to the whole space at $x$ (so actually smooth locally around $x$). For $x$ with $x_1 = 0$, the function value corresponds to the sharp valley in the graph, and we can see that the function is smooth along the valley, and this valley corresponds to the one-dimensional manifold $\{x \mid x_1 = 0\}$ for partial smoothness.

Next, we use the same graph to illustrate the importance of manifold identification. Consider that the red point $x^* = (0, 1.5)$ is the limit point of the iterates of a certain algorithm, and the yellow points and black points are two sequences that both converge to $x^*$. If the iterates of the algorithm are the black points, then clearly except for the limit point itself, all iterates are nonsparse, and thus the final output of the algorithm is also nonsparse unless we can get to exactly the limit point within finite iterations (which is usually impossible for iterative methods). On the other hand, if the iterates are the yellow points, this is the case that the manifold is identified, because all points sit in the valley and enjoy the same sparsity pattern as the limit point $x^*$. This is why we concern about manifold identification when we solve regularized optimization problems.

From this example, we can also see an explanation for why our algorithm with the property of manifold identification performs better than other methods without such a property. Consider a Euclidean space any point $x^*$ with an index set $I$ such that $x_I^* = 0$ and $|I| > 0$. This means that $x^*$ has at least one coordinate being zero, namely $x^*$ contains sparsity. Now let

$$\epsilon_0 := \min_{i \in I^C} \ |x_i^*|,$$

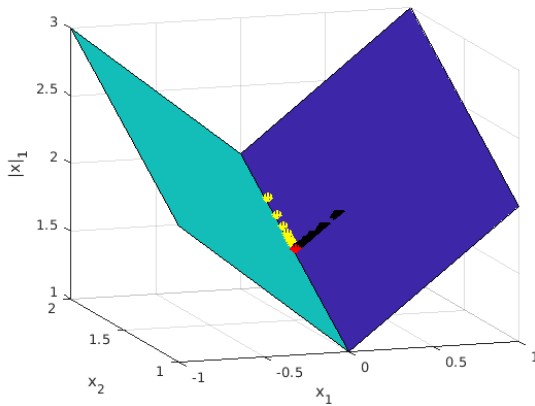

Figure 3: An illustration of partial smoothness of the $\ell_1$ norm.

then from the definition of $I$, $\epsilon_0 > 0$. Fro any sequence $\{x^t\}$ converging to $x^*$, for any $\epsilon \in (0, \epsilon_0)$, we can find $T_\epsilon \geq 0$ such that

$$\left\| x^t - x^* \right\|_2 \leq \epsilon, \quad \forall t \geq T_\epsilon.$$

Therefore, for any $i \notin I$, we must have that $x_i^t \neq 0$ for all $t \geq T_\epsilon$. Otherwise, $\left\| x^t - x^* \right\|_2 \geq \epsilon_0$, but $\epsilon_0 > \epsilon \geq \left\| x^t - x^* \right\|_2$, leading to a contradiction. On the other hand, for any $i \in I$, we can have $x_i^t \neq 0$ for all $t$ without violating the convergence. That being said, for any sequence converging to $x^*$, eventually the iterates cannot be sparser than $x^*$, so the sparsity level of $x^*$, or of its active manifold, is the local upper bound for the sparsity level of points converging to $x^*$. Therefore, if iterates of two algorithms converge to the same limit point, the one with a proven manifold identification ability clearly will produce a higher sparsity level.

Similar to our example here, in applications other than sparsity, iterates converging to a limit point dwell on super-manifolds of the active manifold, and the active manifold is the minimum one that locally describes points with the same structure as the limit point, and thus identifying this manifold is equivalent to finding the locally most ideal structure of the application.

Now back to the sparsity case. One possible concern is the case that the limit point is $(0, 0)$ in the two-dimension example. In this case, the manifold is the 0-dimensional subspace $\{0\}$. If this is the case and manifold identification can be ensured, it means that limit point itself can be found within finite iterations. This case is known as the weak sharp minima (Burke & Ferris, 1993) in nonlinear optimization, and its associated finite termination property is also well-studied.

For this example, We also see that $\|x\|_1$ is partly smooth at any point $x^*$, but the manifold differs with $x^*$. This is a specific benign example, and in other cases, partial smoothness might happen only locally at some points of interest instead of everywhere.

Next, we further extend our argument above to the case of (7). This can be viewed as the $\ell_1$ norm for each group and we can easily obtain similar results. Again, since the group-LASSO norm is also convex and finite everywhere, prox-regularity, regularity, and subdifferential continuity are not issues at all. For the other properties, we consider one group first, then the group-LASSO norm reduces to the $\ell_2$ norm. Clearly, $\|x\|_2$ is smooth locally if $x \neq 0$, with the gradient being $x/\|x\|_2$, but it is nonsmooth at the point $x = 0$, where the subdifferential is the unit ball. This is very similar to the absolute value, whose subdifferential at 0 is the interval $[-1, 1]$. Thus, we can directly apply similar arguments above, and conclude that for any $W^*$, (7) is partly smooth at $W^*$ with respect to the manifold $\mathcal{M}_{W^*} = \{W \mid W_{\mathcal{I}_g} = 0, \forall g : W_{\mathcal{I}_g}^* = 0\}$, which is again a lower-dimensional subspace. Therefore, the manifold of defining the partial smoothness for the group-LASSO norm exactly corresponds to its structured sparsity pattern.

### B.2 Binary Neural Networks

We continue to consider the binary neural network problem. For easier description, for the Euclidean space $\mathcal{E}$ we consider, we will use a vectorized representation for $W, A \in \mathcal{E}$ such that the elements are enumerated as $W_1, \ldots, W_n$ and $\alpha_1, \ldots, \alpha_n$. The corresponding optimization problem can therefore be written as

$$\min_{W,A \in \mathcal{E}} \quad \mathbb{E}_{\xi \sim \mathcal{D}} \left[ f_\xi (W) \right] + \lambda \sum_{i=1}^n \left( \alpha_i \left( w_i + 1 \right)^2 + \left( 1 - \alpha_i \right) \left( w_i - 1 \right)^2 + \delta_{[0,1]} \left( \alpha_i \right) \right), \quad (30)$$

where given any set $C$, $\delta_C$ is the indicator function of $C$, defined as

$$\delta_C(x) = \begin{cases} 0 & \text{if } x \in C, \\ \infty & \text{else.} \end{cases}$$

We see that except for the indicator function part, the objective is smooth, so the real partly smooth term that we treat as the regularizer is

$$\Phi(\alpha) \coloneqq \sum_{i=1}^n \delta_{[0,1]}(\alpha_i).$$

We note that for $\alpha_i \in (0,1)$, the value of $\delta_{[0,1]}(\alpha_i)$ remains a constant zero in a neighborhood of $\alpha_i$, and for $\alpha_i \notin [0,1]$, the indicator function is also constantly infinite within a neighborhood. Thus, the point of nonsmoothness, happens only at $\alpha_i \in \{0,1\}$, and similar to our discussion in the previous subsection, $\Phi$ is partly smooth along directions that we fix those $\alpha_i$ at the boundary (namely, being either 0 or 1) unchanged. The identified manifold therefore corresponds to the entries of $\alpha_i$ that are fixed at 0 or 1, and this can serve as the indicator for the desired binary pattern in this task.

## C Experiment Setting Details

For the weights $w_g$ of each group in (7), for all experiments in Section 5, we follow Deleu & Bengio (2021) to set $w_g = \sqrt{|\mathcal{I}_g|}$. All ProxSSI parameter settings, excluding the regularization weight and the learning rate schedule, follow the default values in their package.

Tables 3 to 13 provide detailed settings of Section 5.2. For the modified VGG19 model, we follow Deleu & Bengio (2021) to eliminate all fully-connected layers except the output layer, and add one batch-norm layer (Ioffe & Szegedy, 2015) after each convolutional layer to simulate modern CNNs like those proposed in He et al. (2016); Huang et al. (2017). For ResNet50 in the structured sparsity experiment in Section 5.2, our version of ResNet50 is the one constructed by the publicly available script at `https://github.com/weiaicunzai/pytorch-cifar100`.

In the unstructured sparsity experiment presented in Section 5.3, for better comparison with existing works in the literature of pruning, we adopt the version of ResNet50 used by Sundar & Dwaraknath (2021).[3] Table 14 provides detailed settings of Section 5.3. For RigL, we use the PyTorch implementation of Sundar & Dwaraknath (2021).

## D More Results from the Experiments

In this section, we provide more details of the results of the experiments we conducted in the main text. In particular, in Fig. 4, we present the change of validation accuracies and group sparsity levels with epochs for the group sparsity tasks in Section 5.2. We then present in Fig. 5 validation accuracies and unstructured sparsity level versus epochs for the task in Section 5.3. We note that although it takes more epochs for RMDA to fully stabilize in terms of manifold identification, the sparsity level usually only changes in a very limited range once (sometimes even before) the validation accuracy becomes steady, meaning that we do not need to run the algorithm for an unreasonably long time to obtain satisfactory results.

---

[3]`https://github.com/varun19299/rigl-reproducibility`.

Table 3: Details of the experimental settings of logistic regression on MNIST in Section 5.2.

| Parameter | Value |
|---|---|
| Data set | MNIST |
| Model | Logistic regression |
| Loss function | Cross entropy |
| Regularization function | Group LASSO |
| Regularization weight | $10^{-3}$ |
| Total epochs | 500 |
| ProxSGD | |
| Learning rate schedule | $\eta(\text{epoch}) = 10^{-1-\lfloor \text{epoch}/50 \rfloor}$ |
| Momentum | $10^{-1}$ |
| ProxSSI | |
| Learning rate schedule | $\eta(\text{epoch}) = 10^{-3-\lfloor \text{epoch}/50 \rfloor}$ |
| RMDA | |
| Restart epochs | $50, 100, 150, 200$ |
| Learning rate schedule | $\eta(\text{epoch}) = \max(10^{-5}, 10^{-1-\lfloor \text{epoch}/50 \rfloor})$ |
| Momentum schedule | $c(\text{epoch}) = \min(1, 10^{-2+\lfloor \text{epoch}/50 \rfloor})$ |

Table 4: Details of the experimental settings of the multi-layer fully-connected NN on FashionMNIST in Section 5.2.

| Parameter | Value |
|---|---|
| Data set | FashionMNIST |
| Model | Fully-connected NN (Table 11) |
| Loss function | Cross entropy |
| Regularization function | Group LASSO |
| Total epochs | 500 |
| ProxSGD | |
| Regularization weight | $10^{-4}$ |
| Learning rate schedule | $\eta(\text{epoch}) = 10^{-1-\lfloor \text{epoch}/50 \rfloor}$ |
| Momentum | $10^{-1}$ |
| ProxSSI | |
| Regularization weight | $4 \times 10^{-6}$ |
| Learning rate schedule | $\eta(\text{epoch}) = 10^{-3-\lfloor \text{epoch}/50 \rfloor}$ |
| RMDA | |
| Regularization weight | $7 \times 10^{-5}$ |
| Restart epochs | $50, 100, 150, 200$ |
| Learning rate schedule | $\eta(\text{epoch}) = \max(10^{-5}, 10^{-1-\lfloor \text{epoch}/50 \rfloor})$ |
| Momentum schedule | $c(\text{epoch}) = \min(1, 10^{-2+\lfloor \text{epoch}/50 \rfloor})$ |

# E  Other Regularizers for Possibly Better Group Sparsity and Generalization

A downside of (7) is that it pushes all groups toward zeros and thus introduces bias in the final model. For its remedy, minimax concave penalty (MCP, Zhang, 2010) is then proposed to penalize only the groups whose norm is smaller than a user-specified threshold. More precisely, given hyperparameters $\lambda \geq 0, \omega \geq 1$, the one-dimensional MCP is defined by

$$
\text{MCP}(w; \lambda, \omega) := \begin{cases} \lambda|w| - \frac{w^2}{2\omega} & \text{if} |w| < \omega\lambda, \\ \frac{\omega\lambda^2}{2} & \text{if} |w| \geq \omega\lambda. \end{cases}
$$

One can then apply the above formulation to the norm of a vector to achieve the effect of inducing group-sparsity. In our case, given an index set $\mathcal{I}_g$ that represents a group, the

Table 5: Details of the experimental settings of LeNet5 on MNIST in Section 5.2

| Parameter | Value |
|---|---|
| Data set | MNIST |
| Model | LeNet5 (Table 12) |
| Loss function | Cross entropy |
| Regularization function | Group LASSO |
| Total epochs | 500 |
| ProxSGD | |
| Regularization weight | $1.2 \times 10^{-4}$ |
| Learning rate schedule | $\eta(\text{epoch}) = 10^{-1-\lfloor \text{epoch}/50 \rfloor}$ |
| Momentum | $10^{-1}$ |
| ProxSSI | |
| Regularization weight | $9 \times 10^{-5}$ |
| Learning rate schedule | $\eta(\text{epoch}) = 10^{-3-\lfloor \text{epoch}/50 \rfloor}$ |
| RMDA | |
| Regularization weight | $10^{-4}$ |
| Restart epochs | $50, 100, 150, 200$ |
| Learning rate schedule | $\eta(\text{epoch}) = \max(10^{-4}, 10^{-\lfloor \text{epoch}/50 \rfloor})$ |
| Momentum schedule | $c(\text{epoch}) = \min(1, 10^{-2+\lfloor \text{epoch}/50 \rfloor})$ |

Table 6: Details of the experimental settings of LeNet5 on FashionMNIST in Section 5.2

| Parameter | Value |
|---|---|
| Data set | FashionMNIST |
| Model | LeNet5 (Table 12) |
| Loss function | Cross entropy |
| Regularization function | Group LASSO |
| Total epochs | 500 |
| ProxSGD | |
| Regularization weight | $1.2 \times 10^{-4}$ |
| Learning rate schedule | $\eta(\text{epoch}) = 10^{-1-\lfloor \text{epoch}/50 \rfloor}$ |
| Momentum | $10^{-1}$ |
| ProxSSI | |
| Regularization weight | $6 \times 10^{-5}$ |
| Learning rate schedule | $\eta(\text{epoch}) = 10^{-3-\lfloor \text{epoch}/50 \rfloor}$ |
| RMDA | |
| Regularization weight | $10^{-4}$ |
| Restart epochs | $50, 100, 150, 200$ |
| Learning rate schedule | $\eta(\text{epoch}) = \max(10^{-4}, 10^{-\lfloor \text{epoch}/50 \rfloor})$ |
| Momentum schedule | $c(\text{epoch}) = \min(1, 10^{-2+\lfloor \text{epoch}/50 \rfloor})$ |

MCP for this group is then computed as (Breheny & Huang, 2009)

$$\text{MCP}\left(W_{\mathcal{I}_g}; \lambda_g, \omega_g\right) := \begin{cases} \lambda_g \left\|W_{\mathcal{I}_g}\right\|^2 - \frac{\left\|W_{\mathcal{I}_g}\right\|^2}{2\omega_g} & \text{if } \left\|W_{\mathcal{I}_g}\right\| < \omega_g \lambda_g, \\ \frac{\omega_g \lambda_g^2}{2} & \text{if} \left\|W_{\mathcal{I}_g}\right\| \geq \omega_g \lambda_g. \end{cases}$$

We then consider

$$\psi(W) = \sum_{g=1}^{|\mathcal{G}|} MCP\left(W_{\mathcal{I}_g}; \lambda_g, \omega_g\right). \tag{31}$$

It is shown in Deleu & Bengio (2021) that group MCP regularization may simultaneously provide higher group sparsity and better validation accuracy than the group LASSO norm in vision and language tasks. Another possibility to enhance sparsity is to add another $\ell_1$-norm or entry-wise MCP regularization to the group-level regularizer. The major drawback

Table 7: Details of the experimental settings of the modified VGG19 on CIFAR10 in Section 5.2.

| Parameter | Value |
|---|---|
| Data set | CIFAR10 |
| Model | VGG19 (Table 13) |
| Loss function | Cross entropy |
| Regularization function | Group LASSO |
| Total epochs | 1000 |
| ProxSGD | |
| Regularization weight | $5 \times 10^{-5}$ |
| Learning rate schedule | $\eta(\text{epoch}) = 10^{-1-\lfloor \text{epoch}/100 \rfloor}$ |
| Momentum | $10^{-1}$ |
| ProxSSI | |
| Regularization weight | $3 \times 10^{-7}$ |
| Learning rate schedule | $\eta(\text{epoch}) = 10^{-3-\lfloor \text{epoch}/100 \rfloor}$ |
| RMDA | |
| Regularization weight | $10^{-4}$ |
| Restart epochs | $100, 200, 300, 400, 500$ |
| Learning rate schedule | $\eta(\text{epoch}) = \max(10^{-6}, 10^{-1-\lfloor \text{epoch}/100 \rfloor})$ |
| Momentum schedule | $c(\text{epoch}) = \min(1, 10^{-2+\lfloor \text{epoch}/100 \rfloor})$ |

Table 8: Details of the experimental settings of the modified VGG19 on CIFAR100 in Section 5.2.

| Parameter | Value |
|---|---|
| Data set | CIFAR100 |
| Model | VGG19 (Table 13) |
| Loss function | Cross entropy |
| Regularization function | Group LASSO |
| Total epochs | 1000 |
| ProxSGD | |
| Regularization weight | $3 \times 10^{-5}$ |
| Learning rate schedule | $\eta(\text{epoch}) = 10^{-1-\lfloor \text{epoch}/100 \rfloor}$ |
| Momentum | $10^{-1}$ |
| ProxSSI | |
| Regularization weight | $10^{-7}$ |
| Learning rate schedule | $\eta(\text{epoch}) = 10^{-3-\lfloor \text{epoch}/100 \rfloor}$ |
| RMDA | |
| Regularization weight | $10^{-4}$ |
| Restart epochs | $100, 200, 300, 400, 500$ |
| Learning rate schedule | $\eta(\text{epoch}) = \max(10^{-6}, 10^{-1-\lfloor \text{epoch}/100 \rfloor})$ |
| Momentum schedule | $c(\text{epoch}) = \min(1, 10^{-2+\lfloor \text{epoch}/100 \rfloor})$ |

of these approaches is the requirement of additional hyperparameters, and we prefer simpler approaches over those with more hyperparameters, as hyperparameter tuning in the latter can be troublesome for users with limited computational resources, and using a simpler setting can also help us to focus on the comparison of the algorithms themselves. The experiment in this subsection is therefore only for illustrating that these more complicated regularizers can be combined with RMDA if the user wishes, and such regularizers might lead to better results. Therefore, we train a version of LeNet5, which is slightly simpler than the one we used in previous experiments, on the MNIST dataset with such regularizers using RMDA and display the respective performance of various regularization schemes in Fig. 6. For the weights $w_g$ of each group in (7), in this experiment we consider the following setting. Let $L_i$ be the collection of all index sets that belong to the $i$-th layer in the network,

Table 9: Details of the experimental settings of ResNet50 on CIFAR10 in Section 5.2. We use the ResNet50 model from the public script `https://github.com/weiaicunzai/pytorch-cifar100`.

| Parameter | Value |
|---|---|
| Data set | CIFAR10 |
| Model | ResNet50 |
| Loss function | Cross entropy |
| Regularization function | Group LASSO |
| Total epochs | 1500 |
| ProxSGD | |
| Regularization weight | $5 \times 10^{-5}$ |
| Learning rate schedule | $\eta(\text{epoch}) = 10^{-1-\lfloor \text{epoch}/150 \rfloor}$ |
| Momentum | $10^{-1}$ |
| ProxSSI | |
| Regularization weight | $3 \times 10^{-7}$ |
| Learning rate schedule | $\eta(\text{epoch}) = 10^{-3-\lfloor \text{epoch}/150 \rfloor}$ |
| RMDA | |
| Regularization weight | $10^{-5}$ |
| Restart epochs | $150, 300, 450, 600$ |
| Learning rate schedule | $\eta(\text{epoch}) = \max(10^{-4}, 10^{-\lfloor \text{epoch}/150 \rfloor})$ |
| Momentum schedule | $c(\text{epoch}) = \min(1, 10^{-2+\lfloor \text{epoch}/150 \rfloor})$ |

Table 10: Details of the experimental settings of ResNet50 on CIFAR100 in Section 5.2. We use the ResNet50 model from the public script `https://github.com/weiaicunzai/pytorch-cifar100`.

| Parameter | Value |
|---|---|
| Data set | CIFAR100 |
| Model | ResNet50 |
| Loss function | Cross entropy |
| Regularization function | Group LASSO |
| Total epochs | 1500 |
| ProxSGD | |
| Regularization weight | $4 \times 10^{-5}$ |
| Learning rate schedule | $\eta(\text{epoch}) = 10^{-1-\lfloor \text{epoch}/150 \rfloor}$ |
| Momentum | $10^{-1}$ |
| ProxSSI | |
| Regularization weight | $3 \times 10^{-7}$ |
| Learning rate schedule | $\eta(\text{epoch}) = 10^{-3-\lfloor \text{epoch}/150 \rfloor}$ |
| RMDA | |
| Regularization weight | $10^{-5}$ |
| Restart epochs | $150, 300, 450, 600$ |
| Learning rate schedule | $\eta(\text{epoch}) = \max(10^{-4}, 10^{-\lfloor \text{epoch}/150 \rfloor})$ |
| Momentum schedule | $c(\text{epoch}) = \min(1, 10^{-2+\lfloor \text{epoch}/150 \rfloor})$ |

and denote

$$N_{L_i} \coloneqq \sum_{\mathcal{I}_j \in L_i} |\mathcal{I}_j|$$

the number of parameters in this layer, for all $i$, we set $w_g = \sqrt{N_{L_i}}$ for all $g$ such that $\mathcal{I}_g \in L_i$. Given two constants $\lambda > 0$ and $,\omega > 1$, The values of $\lambda_g$ and $\omega_g$ in (31) are then assigned as $\lambda_g = \lambda w_g$ and $\omega_g = \omega w_g$.

In this figure, group LASSO is abbreviated as GLASSO; $\ell_1$-norm plus a group LASSO norm, L1GLASSO; group MCP, GMCP; element-wise MCP plus group MCP, L1GMCP. Our results exemplify that different regularization schemes might have different benefits on

Table 11: Details of the multi-layer fully-connected NN. `https://github.com/zihsyuan1214/rmda/blob/master/Experiments/Models/mlp.py`.

| Parameter | Value |
|---|---|
| Type of layers | fully-connected layer |
| Number of layers | 7 |
| Number of output neurons each layer: $1, 2, 3, 4, 5, 6, 7$ | $512, 256, 128, 64, 32, 16, 10$ |
| Activation function for convolution/output layer | relu/softmax |

Table 12: Details of the modified LeNet5 for experiments in Section 5.2. `https://github.com/zihsyuan1214/rmda/blob/master/Experiments/Models/lenet5_large.py`.

| Parameter | Value |
|---|---|
| Number of layers | 4 |
| Number of convolutional layers | 2 |
| Number of fully-connected layers | 2 |
| Size of convolutional kernels | $3 \times 3$ |
| Number of output filters $1, 2$ | $20, 50$ |
| Number of output neurons $3, 4$ | $500, 10$ |
| Kernel size, stride, padding of maxing pooling | $2 \times 2$, none, invalid |
| Operations after convolutional layers | max pooling |
| Activation function for convolution/output layer | relu/softmax |

one of the criteria with proper hyperparameter tuning. The detailed numbers are reported in Table 15 and the experiment settings can be found in Tables 16 and 17.

Table 13: Details of the modified VGG19. `https://github.com/zihsyuan1214/rmda/blob/master/Experiments/Models/vgg19.py`.

| Parameter | Value |
|---|---|
| Number of layers | 17 |
| Number of convolutional layers | 16 |
| Number of fully-connected layers | 1 |
| Size of convolutional kernels | $3 \times 3$ |
| Number of output filters 1-2, 3-4, 5-8, 9-16 | $64, 128, 256, 512$ |
| Kernel size, stride, padding of maxing pooling | $2 \times 2, 2$, invalid |
| Operations after convolutional layers | max pooling, batchnorm |
| Activation function for convolution/output layer | relu/softmax |

Table 14: Details of experimental settings of ResNet50 on CIFAR10 and CIFAR100 for unstructured sparsity. In this experiment, we adopt the version of ResNet50 in Sundar & Dwaraknath (2021).

| Parameter | Value |
|---|---|
| Model | ResNet50 |
| Loss function | Cross entropy |
| RMDA | |
| Data | CIFAR10 |
| Total epochs | 1000 |
| L1 weight | $10^{-5}$ |
| Restart epochs | $150, 300, 450$ |
| Learning rate schedule | $\eta(\text{epoch}) = \max(10^{-3}, 10^{-\lfloor \text{epoch}/150 \rfloor})$ |
| Momentum schedule | $c(\text{epoch}) = \min(1, 10^{-2+\lfloor \text{epoch}/150 \rfloor})$ |
| Data | CIFAR100 |
| Total epochs | 1000 |
| L1 weight | $3 \times 10^{-5}$ |
| Restart epochs | $150, 300, 450$ |
| Learning rate schedule | $\eta(\text{epoch}) = \max(10^{-3}, 10^{-\lfloor \text{epoch}/150 \rfloor})$ |
| Momentum schedule | $c(\text{epoch}) = \min(1, 10^{-2+\lfloor \text{epoch}/150 \rfloor})$ |
| RigL | |
| Data | CIFAR10 |
| Total epochs | 1000 |
| Sparse initialization | erdos-renyi-kernel |
| Density | 0.02 |
| Prune rate | 0.3 |
| Decay schedule | cosine |
| Apply when | step end |
| Interval | 100 |
| End when | 65918 |
| Learning rate | 0.1 |
| Momentum | 0.9 |
| Weight decay | $10^{-4}$ |
| Label smoothing | 0.1 |
| Decay frequency | 20000 |
| Warmup steps | 1760 |
| Decay factor | 0.2 |
| Data | CIFAR100 |
| Total epochs | 1000 |
| Sparse initialization | erdos-renyi-kernel |
| Density | 0.02 |
| Prune rate | 0.3 |
| Decay schedule | cosine |
| Apply when | step end |
| Interval | 100 |
| End when | 65918 |
| Learning rate | 0.1 |
| Momentum | 0.9 |
| Weight decay | $10^{-4}$ |
| Label smoothing | 0.1 |
| Decay frequency | 20000 |
| Warmup steps | 1760 |
| Decay factor | 0.2 |

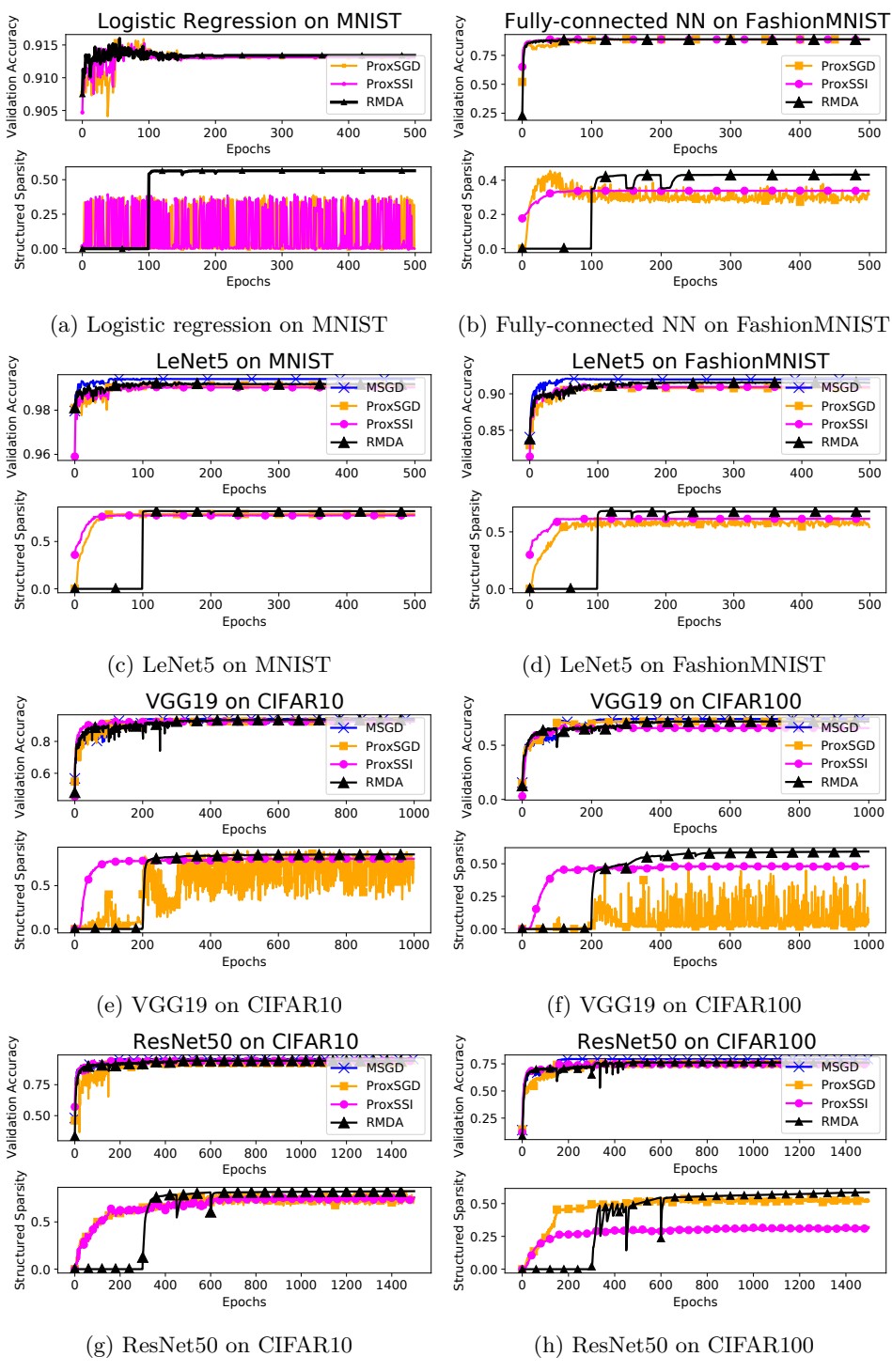

(a) Logistic regression on MNIST

(b) Fully-connected NN on FashionMNIST

(c) LeNet5 on MNIST

(d) LeNet5 on FashionMNIST

(e) VGG19 on CIFAR10

(f) VGG19 on CIFAR100

(g) ResNet50 on CIFAR10

(h) ResNet50 on CIFAR100

Figure 4: Group Sparsity and validation accuracy v.s epochs of different algorithms on various models with the group-LASSO regularization of a single run.

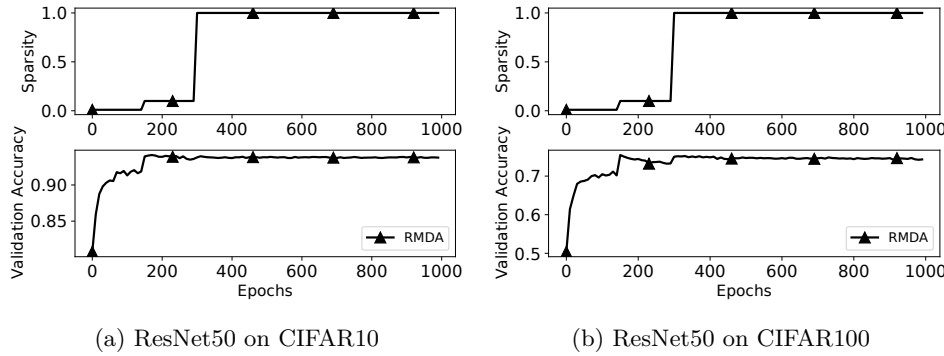

(a) ResNet50 on CIFAR10        (b) ResNet50 on CIFAR100

Figure 5: Unstructured Sparsity and validation accuracy v.s epochs of RMDA on ResNet50 of a single run.

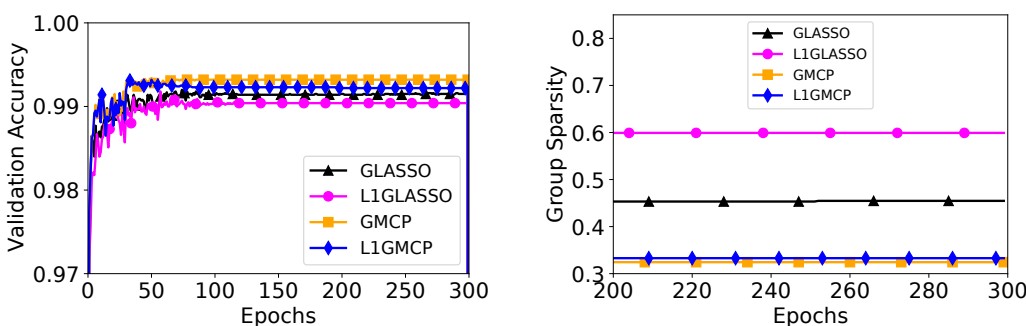

Figure 6: Comparison between group LASSO, L1+group LASSO, Group MCP, L1+Group MCP

Table 15: Results of training LeNet5 on MNIST using RMDA with different regularizers. We report mean and standard deviation of three independent runs.

| Regularizers | Validation accuracy | Group sparsity |
|---|---|---|
| GLASSO | $99.11 \pm 0.06\%$ | $45.33 \pm 0.99\%$ |
| L1GLASSO | $99.02 \pm 0.01\%$ | $58.92 \pm 1.30\%$ |
| GMCP | $99.25 \pm 0.08\%$ | $32.81 \pm 0.96\%$ |
| L1GMCP | $99.21 \pm 0.03\%$ | $32.91 \pm 0.35\%$ |

Table 16: Details of the modified simpler LeNet5 for the experiment in Appendix E. `https://github.com/zihsyuan1214/rmda/blob/master/Experiments/Models/lenet5_small.py`.

| Parameter | Value |
|---|---|
| Number of layers | 5 |
| Number of convolutional layers | 3 |
| Number of fully-connected layers | 2 |
| Size of convolutional kernels | $5 \times 5$ |
| Number of output filters $1, 2$ | $6, 16$ |
| Number of output neurons $3, 4, 5$ | $120, 84, 10$ |
| Kernel size, stride, padding of maxing pooling | $2 \times 2$, none, invalid |
| Operations after convolutional layers | max pooling |
| Activation function for convolution/output layer | relu/softmax |

Table 17: Details of the experimental settings for comparing different regularizers in Appendix E

| Parameter | Value |
| --- | --- |
| Data set | MNIST |
| Model | LeNet5 (Table 16) |
| Loss function | Cross entropy |
| Algorithms | RMDA |
| Total epochs | 300 |
| Restart epochs | $30, 60, 90, 120$ |
| Learning rate schedule | $\eta(\text{epoch}) = \max(10^{-5}, 10^{-1-\lfloor \text{epoch}/30 \rfloor})$ |
| Momentum schedule | $c(\text{epoch}) = \min(1, 10^{-2+\lfloor \text{epoch}/30 \rfloor})$ |
| GLASSO | |
| Group LASSO weight | $10^{-5}$ |
| L1GLASSO | |
| L1 weight | $10^{-4}$ |
| Group LASSO weight | $10^{-5}$ |
| GMCP | |
| Group MCP weight | $10^{-5}$ |
| $\gamma$ | 64 |
| L1GMCP | |
| L1 weight | $10^{-4}$ |
| Group MCP weight | $10^{-5}$ |
| $\gamma$ | 64 |

