# OpenReview forum: "Training Structured Neural Networks Through Manifold Identification and Variance Reduction"
_ICLR.cc/2022/Conference — ICLR 2022 Poster_

### Official Review · Reviewer_KsXr · 2021-11-02

**Correctness:** 1
**Technical Novelty And Significance:** 3
**Empirical Novelty And Significance:** 3
**Recommendation:** 8
**Confidence:** 4

**Main Review:**

### Strengths

1. The paper introduces an algorithm for training neural networks with a regularization term for promoting desired structures in the model. The new method does not incur computation additional to SGD with momentum and could be of practical use. The ideas of the paper are based on well-founded rationales.
2. In general, the paper is quite well-written, with the main ideas outlined clearly. The approach is well-motivated and seems to be built upon established literature. The review of related works is informative.
3. The main theoretical results of the paper (Theorem 1 and Theorem 2) are non-trivial and meaningful, and the proofs are rigorous.

### Weakness

I have a few concerns with the manuscript in its current state.

1. In my opinion, the paper overstates the significance and implications of its results in its presentation. More precisely:
    - In the abstract, introduction, and throughout the discussion, the paper claims that "we rigorously prove that the proposed method can correctly identify within finite steps the underlying structure".
    - In the experiments, they conclude that "RMDA is the only algorithm that steadily identifies the correct structured sparsity".

    I think those are not correct characterizations of the results obtained in this work:
    - Roughly speaking, Theorem 2 for structured sparsity of neural networks just proves that:

         *If the estimator converges to some stationary point W, then its sparsity converges to the sparsity of W within finite step*

    - Similarly, the experiments just show that RMDA has a stable structured sparsity

    While these results are non-trivial, they concern the **stability** of the algorithm, and have little to do with **correctness**: a bad algorithm that got stuck at a non-optimal stationary point also satisfies those properties. To obtain what the paper claims, some more work is needed. In the theory part, results that are equivalent to model-selection consistency for linear models are required. For the simulations part, some experiments with simulated data, when a measure of deviation from the "ground truth" or "correct/optimal sparsity" could be recognized, are necessary.
2. While the results of Section 3 of the manuscript are rigorous, the applications to neural networks (Section 4) lack several important details. When moving from a general framework to a specific case (Section 4.1 and 4.2), two central concepts:
    - the corresponding active manifold,
    - the precise description of the limit point W^*

    are not defined explicitly. This causes significant unnecessary confusion and makes it harder to judge the accuracy and the importance of the results. Specifically,
    - In Definition 1: M is required to be a C^2 manifold, and in general, should be independent of the limit point W^*
    - In Section 4.1 (Structured sparsity), the active manifold seems to be defined as a hyperplane around (and depends on) W^*. To make this fit into the framework, W^* needs to be unique for all random realizations (but this is unlikely). If this is not the case, then M is a union of several hyperplanes of different dimensions (correspond to different limit points for each realization) and is no longer a manifold.

    I believe a clear description of the active manifolds for each case and some verification of their geometric properties would help improve the manuscript.
3. My last concern about the theoretical analysis is its (somewhat hidden and informally defined) assumption that the tentative iterate {W ̃ t} converges almost surely to a certain point W∗. This statement can be interpreted in different ways and needs to be spelled out mathematically. My main question is: Is W* the same for all realizations of randomness? In principle, this is not true, but the analysis was written as if that was the case (see point (2) below).

    The manuscript also stated that verifying this assumption is possible, but these are not the purpose of this work, so they avoid distracting the readers from such technical results. However, it is worth pointing out that W ̃ and W are intertwined in their constructions, and making the convergence of the W~ an assumption is a bit too strong. Even if rigorous results don't need to be provided, some examples and quick discussions about this should be included, perhaps in the appendix. This part will also help resolve the lack of understanding about the structure of W* described above.

### Other comments and questions

- Beginning of Section 3: "In stark contrast to our results, it is actually well-known in convex optimization that those algorithms based on proximal stochastic gradient are *unable to identify the manifold within finite iterations*".
    —> This statement needs more references.

**Summary Of The Paper:**

This paper proposes an algorithm for training neural networks with a regularization term for promoting desired structures in the model. The paper (claims to) prove that the proposed method can correctly identify within finite steps the underlying structure. The simulations then show that the method outperforms existing packages in identifying structured sparsity without compromising prediction accuracy.

**Summary Of The Review:**

Overall, my vote for the paper is a (weak) reject. I think the paper has some strong points and the general results are meaningful. On the other hand, I think the technical writings of Section (Applications to deep learning) are less rigorous and lack important information. Finally, I feel the paper overstates the significance and implications of its results in its presentations.

Update after author's responses and revision: Most of my concerns have been addressed in the revision, with (1) a more rigorous reframing if the significance and importance of the work, and (2) experiments to validate the efficiency of the approach. I think this is a good paper in its current state.

---

> ### Author Response · Authors · 2021-11-09
> **Quick clarifications**
>
> We would like to thank the reviewer for the detailed evaluation of and the invaluable suggestions to our work. Here we would like to provide some quick clarifications, and a more detailed response and revision to address the rest of the comments is in preparation.
>
> 1. For the parts for Sections 3-4, we think the confusion is mainly from a typo in our definition of partial smoothness. In Definition 1 of the submitted manuscript, it is stated that the function is partly smooth at $x^*$ relative to things connected to $W^*$. This is a typo, and $x^*$ should have been $W^*$ instead. Namely, the active manifold is decided by the point $W^*$ that the iterates locate around. Take the L1-norm regularization with 2 dimensions for an example, we know that the absolute value $|x|$ is $C^2$ everywhere except for $x = 0$. Therefore, for any point $z = (z_1, z_2)$, if $z_1 \neq 0$ and  $z_2 = 0$ at a point $z^*$, we see that $\|z\|_1$ is smooth relative to the subspace of $\\{z \mid z_2 = 0\\} $ around $z^*$. Similar arguments apply to the case of $z_1 = 0$, and if $z_1 \neq 0$ and $z_2 \neq 0$, the manifold is the whole space (locally).  Therefore, in this case, the manifold is indeed a linear subspace (with dimension possibly lower than $d-1$ if the original dimension is $d$, so hyperplane is a special case), and this subspace changes for different limit points.
>
> 2. In our analysis, randomness regards the selection of the stochastic gradient, while we didn't involve any randomness in the initial point $W^0$. Our statement of almost surely is therefore related to the stochastic gradients but not the initial point. Indeed it's quite unlikely that the iterates converge almost surely to a single point with all different initializations, even in the case of non-strongly-convex problems such that the set of minimizers is not a singleton. We will make our statement clearer to address this confusion. Here our main focus is indeed that the algorithm finds the sparsity pattern (in the structured-sparsity case) of the limit point in finite steps, while existing methods fail so.
> Thus, the flow of our analysis is:
>
> - Given any initialization, if the iterates converge almost surely (regarding the randomness in the stochastic gradient), the limit point is stationary
>
> - When the limit point in (A) satisfies the nondegeneracy condition and the regularizer is locally partly smooth there with respect to an active manifold, this manifold will be identified in finite iterations.
>
> So what we require is not that all initializations converge to the same limit point, but just that the iterates converge to a point.
>
> 3. We would like to further check with the reviewer about point 1. The reviewer says "results that are equivalent to model-selection consistency for linear models are required," and we do not fully understand this part. Usually model consistency is done on the modeling side but not on the optimization side, and what we intended to claim is the correct structure in the sense of the structure of the stationary point the iterates converging to. Surely a stationary point can be non-optimal, and we can at most show that our algorithm minimizes the objective for convex, or strongly convex, problems (as global minimum for nonconvex optimization is basically not possible, and escaping saddle points for regularized nonconvex problems is still barely studied because without the existence of the Hessian, even charactering saddle points is nontrivial).
> Our plan for the revision is to provide proofs for:
> - The objective converges to the optimum when the problem is convex, or strongly convex.
> - The iterates indeed converge to a point under additional regularity conditions.
>
> Would this address the concerns?
> For the numerical side, we will follow the suggestion to conduct experiments on simulated data to verify the optimality of the identified structure.

---

> > ### Comment · Reviewer_KsXr · 2021-11-18
> > **Further clarifications on the original comment**
> >
> > I want to make some further clarifications based on the response of the authors, mostly about the presentations of their results, as requested by point (3) in the response.
> >
> > One important consideration when study sparsity of neural networks is that the model is highly unidentifiable, and local geometry at different risk-optimal hypothesis may not be the same. Consider just a shallow network with one hidden layer, it is possible (and in fact, common) that there might be two networks: $W_1$, with only 5 non-zero hidden nodes, and $W_2$, with 100 non-zero hidden nodes, that are completely equivalent (in the sense that they produce the same input-output map). As a direct consequence, for any $k$ between 5 and 100, there will also be a network with exactly $k$ non-zero nodes that are equivalent to those networks (in the sense describe above).
> > Now, if we assume further that $W_1$ is irreducible and that noisy data is generated from $W_1$, any methods that claims to be "correct", "optimal", "identify" or "reconstruct" needs to be able to locate a network with optimal sparsity, that is, has only 5 non-zero hidden nodes (thus having an active manifold of dimension 5*(n_in + n_out + 1)). This is not a matter of theoretical or practical considerations, because what good would it does if an algorithm "identifies" a network with 67 nodes and an active manifold of dimension 67*(n_in + n_out +1) in this case?
> >
> > That was what I meant by saying that the result of the work is about *the stability of the algorithm*. What the paper proved for the case of structured sparsity is that, *if the estimator converges to a network with $k$ hidden nodes, then they fall into the active manifold of dimension k*(n_in + n_out +1)*. This has little to do with correctness or optimality (which is a concern raised by another reviewer).

---

> ### Author Response · Authors · 2021-11-20
> **Response (I)**
>
> We would like to thank the reviewer for the detailed evaluation of and the
> invaluable suggestions to our work, and for taking time to engage in discussion with us.
> Our reply to the comments are as follows.
>
>
> > In my opinion, the paper overstates the significance and implications of its results in its presentation...
> > While these results are non-trivial, they concern the stability of the
> > algorithm, and have little to do with correctness: a bad algorithm that got
> > stuck at a non-optimal stationary point also satisfies those properties. To
> > obtain what the paper claims, some more work is needed. In the theory part,
> > results that are equivalent to model-selection consistency for linear models
> > are required. For the simulations part, some experiments with simulated data,
> > when a measure of deviation from the "ground truth" or "correct/optimal
> > sparsity" could be recognized, are necessary.
>
> We totally agree with the reviewer and thank you for pointing out the issue of
> saying correct or optimal.
> As summarized, our intention is "locally optimal", or correct as the same
> structure at the asymptotic point of convergence (which is the one the iterates
> hope to capture, given its convergence to this point), and we are sorry about this misunderstanding.
> We have updated the descriptions to use different wordings like stationary
> structure, or the structure at the limit point, to better communicate what we
> mean, and we hope this is satisfactory.
> As you pointed out, global optimal is almost impossible to achieve, but we hope
> our description in the summary also convinces you that this a little bit more
> than just stability, and actually finds the best an algorithm can given
> convergence to the same point.
> Finding a limit point with dimension 67*(n_in + n_out +1) might not be very
> meaningful if the global ground truth is 5, but at least in the situation that
> the limit point is indeed 67 *(n_in + n_out + 1), finding exactly this
> structure is at least better than converging to this point but the output is
> with dimension 100 * (n_in + n_out + 1), or some arbitrarily number.
> So we think this is still of its meaning, and the numerical results also
> support that finding the active manifold can be of use to improve results.
>
> > For the simulations part, some experiments with simulated data, when a
> > measure of deviation from the "ground truth" or "correct/optimal sparsity"
> > could be recognized, are necessary.
>
> We would like to thank you for the suggestion of this experiment.
> Now this experiment has been added, and we think it indeed provides further
> information about confirming the difference in manifold identification of the
> compared methods.
>
> > While the results of Section 3 of the manuscript are rigorous, the
> >applications to neural networks (Section 4) lack several important details.
> >When moving from a general framework to a specific case (Section 4.1 and 4.2), two central concepts:
>      >- the corresponding active manifold,
>      >- the precise description of the limit point W^*
> >
> >are not defined explicitly. This causes significant unnecessary confusion and
> >makes it harder to judge the accuracy and the importance of the results.
> >Specifically,
>      >- In Definition 1: M is required to be a C^2 manifold, and in general, should be independent of the limit point W^*
>      >- In Section 4.1 (Structured sparsity), the active manifold seems to be defined as a hyperplane around (and depends on) W^*. To make this fit into
> the framework, W^* needs to be unique for all random realizations (but this is unlikely). If this is not the case, then M is a union of several hyperplanes of different dimensions (correspond to different limit points for each realization) and is no longer a manifold.
>
> As replied previously, we think this was mainly due to a typo in the definition, and from your reply, we hope this issue is now cleared.
>
> > I believe a clear description of the active manifolds for each case and some
> > verification of their geometric properties would help improve the manuscript.
>
> We would like to thank you for this suggestion that led to our example to argue
> that manifold identification helps to find the locally best structure.
> As summarized, now we have followed your suggestion to discuss more details
> (most on the sparsity one) in Appendix C. The introduction has also been
> updated to articulate this concept with a toy example.

---

> > ### Author Response · Authors · 2021-11-20
> > **Response (II)**
> >
> >
> > > My last concern about the theoretical analysis is its (somewhat hidden and
> > > informally defined) assumption that the tentative iterate {W ̃ t} converges
> > > almost surely to a certain point W∗. This statement can be interpreted in
> > > different ways and needs to be spelled out mathematically. My main question
> > > is: Is W* the same for all realizations of randomness? In principle, this is
> > > not true, but the analysis was written as if that was the case (see point (2)
> > > below).
> >
> > > The manuscript also stated that verifying this assumption is possible, but
> > > these are not the purpose of this work, so they avoid distracting the readers
> > > from such technical results. However, it is worth pointing out that W ̃ and W
> > > are intertwined in their constructions, and making the convergence of the W~
> > > an assumption is a bit too strong. Even if rigorous results don't need to be
> > > provided, some examples and quick discussions about this should be included,
> > > perhaps in the appendix. This part will also help resolve the lack of
> > > understanding about the structure of W* described above.
> >
> > Thank you for spotting the caveats in our previous analysis and description.
> > Now we have updated the theoretical results and the description.
> > Basically, now we mention explicitly that given any $W^0$, and separate the
> > nonzero-probability events of converging to different $W^*$.
> > We hope this updated description now removes the possible mathematical
> > ambiguity, and the assumption should now be less strong, as indeed for all
> > cases to converge to the same $W^*$ is too strong, and actually unrealistic, an
> > assumption.
> >
> > > Beginning of Section 3: "In stark contrast to our results, it is actually
> > > well-known in convex optimization that those algorithms based on proximal
> > > stochastic gradient are unable to identify the manifold within finite
> > > iterations". —> This statement needs more references.
> >
> > As also suggested by Reviewer mrfp, we have now updated this description to
> > make it clearer, and also added references per your suggestion.

---

> ### Author Response · Authors · 2021-11-28
> **Any further concerns?**
>
> We hope the revision has addressed most concerns of the reviewer raised in evaluating our initial submission. If the reviewer has any further concerns of or comments on the revision, we are more than happy to answer them. However, as the review period is nearing its end, we hope we can get a chance to address such further concerns, if any, while we are still allowed to discuss here.

---

> > ### Comment · Reviewer_KsXr · 2021-11-28
> > **Score update**
> >
> > Thanks for your responses and your careful revision of the paper. I'm happy with the revision and will increase the score.

---

### Official Review · Reviewer_MELK · 2021-11-02

**Correctness:** 4
**Technical Novelty And Significance:** 3
**Empirical Novelty And Significance:** 3
**Recommendation:** 8
**Confidence:** 3

**Main Review:**

Positive aspects:

1. The proposed algorithm is well motivated.
2. The proposed algorithm is a unique and different take on structured learning.
3. The proposed algorithm and the related math is well explained in the paper.


Negative aspects:
1. Experiments are limited. It is understandable that the main goal of the paper is to share a new proximal based method for structured learning but it will be good to have a few more experiments.
2. It will also be good if comparisons with other structure learning-based methods are done.
3. Related work is missing a recent work that leverages proximal operators to find a sparse network from a pretrained network which can also be applied during training for group sparsity. Verma, Sagar, and Pesquet, Jean-Christophe. "Sparsifying Networks via Subdifferential Inclusion."  International Conference on Machine Learning. PMLR, 2021.

**Summary Of The Paper:**

In this paper, the authors propose Regularized Modernized Dual Averaging (RDMA) algorithm to train structured neural networks. This is a proximal method that achieves variance reduction without any extra cost per iteration. The algorithm does not require any extra hyperparameters than what stochastic gradient descent requires. Authors theoretically prove that structure identification is guaranteed after a finite number of iterations.

Experiments have been performed to obtain group sparse neural networks. Networks and datasets used in the experiments are simple logistic regression network on MNIST, 7-layer fully-connected network on FashionMNIST, LeNet5 on MNIST, and VGG16 on CIFAR10. The method has been compared with SGD, ProxSGD, and ProxSSI.

**Summary Of The Review:**

I am leaning towards acceptance given that authors address the problems listed out. Overall the idea is good and the paper is well written to be considered at par with ICLR.

---

> ### Author Response · Authors · 2021-11-20
> **Reply**
>
> We would like to thank the reviewer for the careful evaluation and the
> suggestion of additional experiments and pointing us to a related work.
> Our response to the comments are as follows.
>
> 1. In the revision, we have added discussion about pruning methods including
> the paper suggested by the reviewer.  However, we respectfully disagree with
> the reviewer about that the work of Verma & Jean-Christophe (2021) can be
> incorporated in the training stage, or at least it will be extremely expensive.
> For example, if we periodically prune the dense network after certain epochs of
> training without sparsity-inducing regularization, then:
>
>     - The pruning part itself is also iterative and involves many epochs, so the
>   running time grows significantly if we need to do pruning multiple times
>     - After pruning, if we maintain the same sparsity pattern in subsequent epochs,
>   because the previously pruned network might not be a good one, especially at
>   the early stage of training, we might wrongly prune out groups that are
>   actually useful (which is disastrous when happened at the input layer).
>   But if we do not keep the same sparsity pattern, pruning in the training
>   stage becomes meaningless.
>   On the other hand, regularized training allows change of the sparsity pattern
>   across epochs and can avoid this issue.
>
> Therefore, we did not further investigate the possibility of pruning applied at
> the training stage.
>
> 2. We have added more experiments as described in the summary, and all
> experiments showed that RMDA is consistently better.
> We hope this addresses the concern of limited amount of experiments.
> In the additional experiments, we have also followed your suggestion to compare
> with another structure-learning-based method. In particular, we considered
> pruning, following your mentioning of the related work.
> We were unable to compare with the method in Verma & Jean-Christophe (2021)
> itself numerically though, as their code is not publicly available, and we
> therefore used a close competitor, RigL, that the mentioned work compared
> with.
> Our result shows a clear advantage of RMDA over RigL in terms of the validation
> accuracy when the sparsity levels are the same (or when RMDA is slightly
> sparser).
> For the CIFAR100 case, RMDA shows a significant advantage over RigL.
> We have also included our code for this experiment in the supplementary
> materials, in case there is any concern of unfair comparison caused by any
> accident.

---

### Official Review · Reviewer_mrfp · 2021-11-08

**Correctness:** 3
**Technical Novelty And Significance:** 3
**Empirical Novelty And Significance:** 2
**Recommendation:** 6
**Confidence:** 4

**Main Review:**

Using momentum and different regularizers to impose the desired structure during optimization and esp. NN training is common in machine learning. The proposed RMDA can be viewed as a non-trivial extension of the dial averaging algorithm to use momentum.

1. I found the claims not 100% accurate, and sometimes misleading:
 - **Variance reduction beyond ...**: It was not clear how this achieved? can the authors explain what they meant by variance reduction and where in the paper they showed that? Also, despite claiming that RDMA's cost is the same as SGD, keeping track of momentum and dual averaging makes it slightly more complex than SGD (although still better than other variance reduction techniques)
- **Guaranteed strucutre identification**: I find this slightly exaggeration of the results as there is no theoretical result on the convergence of the RDMA at all. Theorem 2 assumes that the algorithm converges, and under some assumptions, the converged parameter belongs to the active manifold.
- **Superior empirical performance**: "RDMA identifies the optimum structure" is hard to prove, as there is no way to show that the structure found by any algorithm is the best and optimum, even though it is better than baseline.

2. The theoretical analysis is not complete, and is based on the restrictive assumptions and expecting that with the given values of $\\beta_t$, $\\eta_t$ the algorithm converges.
3. In theorem 2, does $T_0$ refer to the extra iterations of RDMA after having $\\tilde{W}^t$ converged? or it includes all iterations of RDMA, including the convergence of $\\tilde{W}^t$?
4. Section 3, It is stated that in convex optimization, "algorithms based on proximal stochastic gradient are unable to identify the manifold within finite iterations". However, RMDA can identify the active manifold in finite iterations. What is the main reason to achieve this contradicting result? Is it because of the set of assumptions made in the analysis? or the way that momentum is incorporated in developing the algorithm?

minor suggestions:
- It would be much better to define all notations used in the paper, for example the notations for subdifferential, interior, relative-interior (although mostly standard).


**Summary Of The Paper:**

Regularization is generally used to impose desired structure on NN during training. The authors developed RMDA (Regularized Modernized Dual Averaging) which uses the weighted average of the previous SG to compute the tentative update via proximal operation associated with the regularizer. Then the parameters of the model are updated in the direction of this tentative update with a pre-defined factor. The authors theoretically analyzed the performance of their algorithm and showed that under some assumptions, RMDA can identify the structure of model in finite iterations.


**Summary Of The Review:**

By adding momentum and proximal operation associated with the regularization to the modernized dual averaging algorithm, the authors developed RDMA. Using tools from non-linear optimization and manifold identification, they theoretically showed that if RDMA converges, then it will find the active manifold after a finite number of iterations. Experimentally, they showed that RDMA outperforms proxSGD and ProxSSI in terms of both accuracy (sometimes) and group sparsity (always).

---

> ### Author Response · Authors · 2021-11-09
> **Quick Clarifications**
>
> We would like to thank the reviewer for the detailed evaluation of and the invaluable suggestions to our work. Here we would like to provide some quick clarifications, and a more detailed response and revision to address the rest of the comments is in preparation.
>
> 1. Variance reduction:
> - In the contribution summary, we mean that the variance of the gradient estimator constructed decreases to zero, just like other variance reduction methods. In our algorithm, the estimator is $\alpha_t^{-1} V^t$ as that is the linear term we used in the subproblem, which is where one would use the true gradient if available. We have shown in Lemma 1 that this is achieved because the estimator converges to the true gradient $\\nabla f(W^t)$ almost surely.
> Since the individual stochastic gradients have bounded variance and whenever $W^t$ stays within a bounded region, $\\nabla f(W^t)$ is also upper-bounded, we get from the dominated convergence theorem that almost sure convergence implies convergence in $L_2$ as well. This means that the variance  $\\mathbb{E}[\\|\\alpha_t^{-1} V^t - \\nabla f(W^t)\\|^2]$ also decreases to zero. This is how RMDA achieves variance reduction.
>
> - Our description in Section 3 means that for plain proximal stochastic gradient methods, so those without the mechanism of variance reduction. When variance reduction is used, identifying the active manifold is possible. This is why RMDA achieves manifold identification while existing approaches like proxSGD and proxSSI don't.
>
>
> 2. Convergence and manifold identification:
> - Our convergence result for the algorithm is in Theorem 1: our result there shows that if the iterates converge to a point, it will be a stationary one. To guarantee the assumption that there is only one limit point requires some more regularity conditions indeed, even in the convex case and even without the mixing with the previous iterate. We will soon add an analysis for convergence of the iterates under additional regularity conditions in the coming revision.
>
> - Moreover, the assumed conditions are all just for generalization to include more applications. At least in the case of the group-LASSO norm and the MCP penalty for which we have conducted experiments, the requirements for the regularizer all hold true. The nondegeneracy condition indeed cannot be verified a priori, but since the volume of the relative boundary of $\partial F(W^*)$ is zero, degenerate problems are barely faced in practice, and can be easily avoided by perturbing just one data point slightly.
>
> - Our algorithm doesn't require any extra iterations/epochs except for the procedure described, so $T_0$ in Theorem 2 doesn't mean any extra iterations. We mean after a certain number of epochs of RMDA, the manifold of the limit point will be identified. The identification is like a by-product of the main algorithm that doesn't require any extra effort.
>
>
> 3. Superior empirical performance: By optimum and best we mean in the optimization sense, namely the structure is the same as that of a (local) minimizer. We agree with the reviewer that indeed this is hard to verify numerically in the current setting, unless one knows the ground truth a priori (which is why we also provided theoretical results), and we should have stated it clearer. Following the suggestion of reviewer KsXr, we will conduct an experiment with simulated data to verify this more rigorously, as in this case we can easily check whether the current model has the same sparsity pattern as the ground truth.

---

> ### Author Response · Authors · 2021-11-20
> **Response (I)**
>
> We would like to thank the reviewer for the detailed evaluation of and the
> invaluable suggestions to our work.
> Our reply to the comments are as follows.
>
>
> > **Variance reduction beyond ...**: It was not clear how this achieved? Can the
> > authors explain what they meant by variance reduction and where in the paper
> > they showed that? Also, despite claiming that RDMA's cost is the same as SGD,
> > keeping track of momentum and dual averaging makes it slightly more complex
> > than SGD (although still better than other variance reduction techniques)
>
> Now we have added a new result in Lemma 2 to show directly that the variance of
> $\\alpha_t^{-1} V_t$, as an estimator for $\\nabla f(W^{t-1})$, converges to zero.
> The original result also implies that since $\\alpha_t^{-1} V_t$ converges to
> $\\nabla f(W^{t-1})$, the variance of it decreases to zero.
> We have also added some remarks before Lemma 2 to clarify what we meant by
> variance reduction.
> A pointer to Lemma 2 is now also given in the contribution summary when we
> mentioned variance reduction.
>
> For the cost part, we intended to mean proximal SGD with momentum (in this
> case, the cost, including vector operations, is really exactly the same), and
> thanks for pointing this ambiguity out.
> We also note that even when comparing with momentum SGD without the proximal
> operation, our reported running time per epoch for VGG16 (also appeared in the
> original version) suggested that indeed the real time cost of RMDA is almost
> the same.
>
> > **Guaranteed structure identification**: I find this slightly exaggeration of
> > the results as there is no theoretical result on the convergence of the RDMA
> > at all. Theorem 2 assumes that the algorithm converges, and under some
> > assumptions, the converged parameter belongs to the active manifold.
> > The theoretical analysis is not complete, and is based on the restrictive
> > assumptions and expecting that with the given values of $\\beta_t, \\eta_t$,
> > the algorithm converges.
>
> As discussed previously, our original Theorem 1 (now Theorem 2) is the
> convergence result.
> The assumption is only that the iterates converge to a point, but we didn't
> assume the point of convergence is stationary. Instead, Theorem 2 proved it.
>
> But to further address this concern, now we have added one more convergence
> result in Theorem 1 for RMDA, and we hope this new result also helps to
> establish more theoretical supports for RMDA.
>
> > **Superior empirical performance**: "RDMA identifies the optimum structure" is
> > hard to prove, as there is no way to show that the structure found by any
> > algorithm is the best and optimum, even though it is better than baseline.
>
> We agree that indeed optimal is not the right word to describe what we do, and
> have updated our descriptions.
> However, following the experiment suggested by Reviewer KsXr, we now have an
> experiment of synthetic data, and we can know what the best and optimal
> structure is in this case. And indeed our new numerical results in this case
> show that the optimal structure can indeed be found (in some special settings)
> by RMDA, while other methods, although all converging to the global optimal
> point, do not find this structure.
>
> > In theorem 2, does $T_0$ refer to the extra iterations of RDMA after having
> > $\\tilde W^t$ converged? Or it includes all iterations of RDMA, including the
> > convergence of $\\tilde W^t$?
>
> As described in our previous reply, there is no additional iterations in RMDA,
> and identification is spontaneous, simply that after some iterations $T_0$, all
> later iterates possess the same structure as the asymptotic limit point.
>
> We also think that there might be a misunderstanding.
> There is no way to clearly define or decide that a sequence "has converged", as
> converging is a gradual process, and reaching the point of convergence can only
> happen asymptotically, when $t \\rightarrow \\infty$.
> If the reviewer meant that after the validation accuracy or the training
> objective stops improving, there is no way to tell if manifold identification
> happens before or after the former (as validation is something not directly
> connected with optimization), but for the latter, as we can see from the
> illustrated example in the revision, when the manifold is identified, usually
> the objective also experiences some drop simultaneously (to descent to the
> valley in the case of our graphical example).

---

> > ### Author Response · Authors · 2021-11-20
> > **Response (II)**
> >
> >
> > > Section 3, It is stated that in convex optimization, "algorithms based on
> > > proximal stochastic gradient are unable to identify the manifold within
> > > finite iterations". However, RMDA can identify the active manifold in finite
> > > iterations. What is the main reason to achieve this contradicting result? Is
> > > it because of the set of assumptions made in the analysis? Or the way that
> > > momentum is incorporated in developing the algorithm?
> >
> > We intended to refer to plain proximal SGD (possibly with momentum), without
> > any mechanism of variance reduction, and have updated our text to reflect this
> > (and added references for it, as also suggested by Reviewer KsXr), so this is
> > now not contradicting, and thank you for pointing this unclear part in our
> > writing out.
> > The difference is from the way RMDA accumulates and uses momentum that is
> > different from that of proximal momentum SGD, and this is not just the
> > assumptions or artifacts in theory, this is also observed in the numerical
> > experiments that other methods in comparison cannot identify an active
> > manifold even if they are really converging to a limit point.
> >
> > > It would be much better to define all notations used in the paper, for
> > > example the notations for subdifferential, interior, relative-interior
> > > (although mostly standard).
> >
> > Now those are added in Section 3 before we introduce partial smoothness.

---

### Author Response · Authors · 2021-11-23
**Latest updates**

We would like to thank all reviewers again for their time in engaging in discussions with us and for the evaluation of the revisions.
As suggested by Reviewers mrfp and MELK, we have improved the visualization of the figures. The new version now should be hopefully more legible. We have also fixed the page format issue in the reference page raised by Reviewer MELK.

---

### Decision · Program_Chairs · 2022-01-20

**Decision:**

Accept (Poster)

**Comment:**

The paper develops optimization algorithms for fitting structured neural networks. It focuses on the manifold identification property, which guarantees after finitely many iterations, all iterates have the same sparsity structure as at convergence. The proposed method extends dual averaging to include momentum. The paper’s analysis shows that if the proposed method converges, it converges to a stationary point, and identifies the sparsity pattern of the limit in finitely many iterations. Experiments show improvements in sparsification compared to existing two-step sparsifiers, without a degradation in accuracy.

The initial review raised concerns about clarity, as well as some of the claimed significance of the results: the paper does not prove that an optimal, or even good sparsity structure is obtained — rather, it proves that the sparsity structure at convergence is obtained after finitely many iterations. The reviewers also raised a number of detailed concerns about the paper’s mathematical exposition.

After considering the authors response, and a revision which significantly clarified both the paper’s notation and its main claims, the reviewers converged to a recommendation to accept. The paper provides a principled approach to sparsification, with supporting theory (albeit about finite identification, rather than optimality). The proposed algorithm appears quite practical and is supported by experiments demonstrating improvements over existing sparsification methods.